

# Spatio-temporal dynamics of sediment transfer systems in landslide-prone alpine catchments

François Clapuyt[1], Veerle Vanacker[1], Fritz Schlunegger[2], Marcus Christl[3], Kristof Van Oost[1]

[1] Earth and Life Institute, Georges Lemaître Centre for Earth and Climate Research, Université Catholique de Louvain, Place
Pasteur, 3 Bte L4.03.08, 1348 Louvain-la-Neuve, Belgium.
[2] Institut für Geologie, Universität Bern, Institut für Geologie, Baltzerstrasse 1+3, 3012 Bern, Switzerland.
[3] Laboratory of Ion Beam Physics, ETH Zurich, Otto-Stern-Weg 5, 8093 Zürich, Switzerland.

*Correspondence to:* François Clapuyt (francois.clapuyt@uclouvain.be).





**Abstract.** Tectonic and geomorphic processes drive landscape evolution over different spatial and temporal scales. In mountainous environments, river incision sets the pace of landscape evolution, and hillslopes respond to channel incision by e.g. gully retreat, bank erosion and landslides. Sediment produced during stochastic landslide events leads to mobilisation of soil and regolith on the slopes that can later be transported by gravity and water to the river network. Quantifying sediment storage and conveyance requires an integrated approach accounting for different space and time scales. To better understand mechanisms and spatial and temporal scales of geomorphic connectivity in mountainous environments, we characterised the sediment cascade of the Entle River catchment located in the foothills of the Central Swiss Alps. We quantified sediment fluxes over annual, decadal and millennial time scales using respectively UAV-SfM techniques, classic photogrammetry and in situ produced cosmogenic radionuclides. At the annual scale (2013-2015), the sediment budget of the Schimbrig earthflow is roughly in equilibrium, despite the fact that we measured intense sediment redistribution on the hillslopes. At the decadal scale (1962-1998), Schwab et al. (2008) reported episodes of sediment export that were not directly related to increased geomorphic activity on the hillslopes. At the millennial scale, catchment-wide denudation rates show a positive relationship with downstream distance or drainage area, when ignoring landslide-affected catchments. The latter are characterised by a negative relationship between denudation rates and downstream distance, along with high variability in denudation rates. The high denudation rates that we measured in the earthflow-affected Schimbrig catchment are illustrative for its high rates of geomorphic activity in comparison to adjacent areas. Our data show that the elevated denudation rates of the landslide-affected catchments are not necessary traceable when analyzing long-term sediment fluxes of the wider geographic area, as the landslide-affected catchments are often only a small fraction of the total catchment. The multi-temporal assessment of sediment fluxes indicates that (1) landslides can provide local sediment pulses, and mobilise material that becomes available for further mobilisation and transport when hillslopes and channels are connected. (2) Connection and disconnection cycles occur at decadal time scale. (3) Phases of high geomorphic activity at the catchment scale are episodic over thousands of years. Consequently, one single landslide has not necessarily an impact on the long-term sediment budget of first-order catchments. Rather, it is the cumulated effect of multiple landslides which are intermittently connected to the channel network at the decadal scale that may regulate sediment fluxes at the regional scale over the millennial time scale.

## 1 Introduction

The segmentation of the sediment pathway into distinct cascades is a widely used concept to describe the routing of sediment particles from sources to sinks throughout a landscape (Walling, 1983). Among other factors, e.g. topography, lithology, climate or tectonic activity (e.g. Aalto et al., 2006; Montgomery and Brandon, 2002; Whipple and Tucker, 1999), the geomorphic coupling and sediment connectivity control the efficiency of how sediment is transferred in geomorphic systems, and they eventually condition the pace at which landscapes evolve through time (Bracken et al., 2015; Fryirs, 2013; Harvey, 2001; Heckmann and Schwanghart, 2013). The *geomorphic coupling* between distinct landscape elements is commonly seen as a measure of how individual landforms are linked through sediment transport (e.g. Harvey, 2001; Heckmann and





Schwanghart, 2013), while the term *sediment connectivity* has been employed for characterizing the transfer of sediments at a larger scale, which includes the potential sources and sinks within a geomorphic system (Bracken et al., 2015). Accordingly, a large connectivity requires an implicit geomorphic coupling between distinct landscape units (Bracken et al., 2015).

In this context, most research has focused on how the connectivity between landslides and trunk channels influences the overall sediment budget of a landscape. Because landslide processes are a dominant source of sediments in mountainous environments (Korup et al., 2010), one would expect that the magnitude and frequency of landsliding (e.g. Crozier and Glade, 1999; Hovius et al., 1997; Malamud et al., 2004) would directly impact the bulk sediment flux of a drainage basin. Nevertheless, the contribution of landslides to the overall sediment budget is still poorly constrained, because their geomorphic efficiency varies according to the mechanisms and scales of sediment connectivity (Benda and Dunne, 1997; Bennett et al., 2014), and since landslides stochastically supply sediment to the river network. Furthermore, field studies have shown that the landscape buffering capacity can strongly vary from several years (Berger et al., 2011; Fuller and Marden, 2010) to decades (e.g. Bennett et al., 2013; Schwab et al., 2008) and millennia (e.g. Wang et al., 2017). However, the inherent stochastic nature of sediment production and transport through landsliding prohibits a linear upscaling of small-to-medium scale geomorphic process assessments, as well as extraction of a particular erosion mechanism from the entire sediment cascade using long-term/large-scale methods (Bennett et al., 2014; Bracken et al., 2015).

Few studies have attempted to unravel spatio-temporal patterns of the sediment cascade in landslide-affected catchments. Mackey et al. (2009) compared decadal measurements of surface displacement velocities of an earthflow derived from historical airphotos with millennial transport rates for the transport zone, derived from meteoric [10]Be inventories. Their study highlighted that the displacement rate of the Eel earthflow (northern California) was highly episodic in time, as the earthflow acted as a source of sediments over the last 150 years with an erosion rate that was more than 20 times faster than the millennial sediment transport rate. In a similar study, Delong et al. (2012) measured surface displacement rates of the Mill Gulch earthflow from light detection and ranging (LiDAR) data and compared this data with [10]Be-derived denudation rates of two adjacent catchments. These authors reported, short-term denudation rates (2003 and 2007) that were similar to long-term ones. Accordingly, a spatio-temporal approach is needed to assess the sediment dynamics and the mechanisms of sediment connectivity in landslide-prone environments, which is the focus of this paper.

Here, we quantify the spatio-temporal dynamics of sediment transfer systems, and elucidate the processes controlling hillslope-channel coupling over $10^0$-$10^3$ years. Integrating sediment dynamics in landslide-prone environments over different time scales enables us to capture the stochastic character of geomorphic processes, e.g. landslides, and assess the propagation of sediment pulses from the hillslopes to the channels. Our analysis is based on a multi-temporal assessment of sediment dynamics in a small mountainous catchment. To this extent, we establish a dataset of denudation rates and sediment fluxes. At the annual scale, the geomorphic processes in the hillslope domain ($10^0$-$10^1$ km$^2$) were monitored using very high-resolution topographic reconstructions. At the decadal scale, the sediment dynamics of first-order catchments were quantified from time-series of digital elevation models using classic photogrammetry. Finally, catchment-averaged cosmogenically radionuclide (CRN)-derived processes rates integrated geomorphic process rates over the millennial time scale. This framework was applied to the





Entle River catchment where the Schimbrig earthflow is located. We analysed the sediment cascade in terms of spatio-temporal patterns in denudation rates (L T$^{-1}$), and sediment fluxes (L$^3$ T$^{-1}$). The catchment-scale denudation rate quantifies the surface lowering per unit of time (L T$^{-1}$) and is scale-invariant. The sediment flux (L$^3$ T$^{-1}$) is the volume of sediment material exported or evacuated from a given area per unit of time, and quantifies the rate of sediment transfer between landscape units, i.e.

between hillslopes and channels. In addition, it enables us to track the internal sediment dynamics within a given catchment. Following these concepts, we first quantified millennial-scale sediment fluxes from the Schimbrig catchment. Then, we discuss the spatial variability in millennial denudation rates for the Entlebuch region, by integrating our new [10]Be-derived denudation rates with previously published data on the Entle catchment by Van den Berg et al. (2012). Finally, we analyze the temporal variability in sediment fluxes, by comparing the millenial scale sediment fluxes with decadal (Schwab et al., 2008) and annual

sediment flux data (Clapuyt et al., 2017).

## 2 Material and methods

### 2.1 The Entle and the Schimbrig catchments

The Entle catchment is a 64 km$^2$ river catchment located in the northern foothills of the Central Swiss Alps, between Bern and Lucerne. Its elevation is ranging between 680 m at the outlet located near the Entlebuch village and 1,815 m a.s.l. on top of

the Schimbrig summit. The study area lies on the intersection between the Swiss Plateau, i.e. the Molasse Basin, and the frontal thrusts of the Alpine orogeny (Schlunegger et al., 2016a). The Molasse unit, covering the lower reaches of the catchment, is composed of Late Oligocene conglomerate bedrock knobs, forming erosion-resistant low ridges. The intermediate part of the catchment is covered by the Subalpine Flysch, while the higher SW-NE-oriented ridge is composed of Cretaceous carbonate rocks of the Helvetic thrust sheet (Figure 1).

The Entle catchment is dissected by a 7 km-long central inner gorge with two tributaries, i.e. the Grosse and the Kleine Entle, that are deeply incised into a more than 100 m thick unconsolidated glacial till. The inner gorge contains knickzones in its longitudinal profile, and several cut terraces are visible. A [10]Be-based sediment budget, which covers the last ca. 2,000 years, highlighted that erosion rates in the inner gorge are more than 4 times higher than in the non-incised reaches over (Van den Berg et al., 2012). Therefore, the river system can be considered as supply-limited, and eroded sediments have been efficiently

transported to downstream reaches. Lateral and terminal moraines deposited by the Entle glacier during the Last Glacial Maximum (LGM) are present on both sides of the inner gorge, and in the headwaters of the Grosse and Kleine Entle rivers (Figure 1). The glacial till was deposited during repetitive and extensive glaciations during the Pleistocene. Sidewalls of river valleys are subject to widespread rotational and translational landslides. Flysch-covered areas are mainly affected by landslide processes categorized as earthflows.





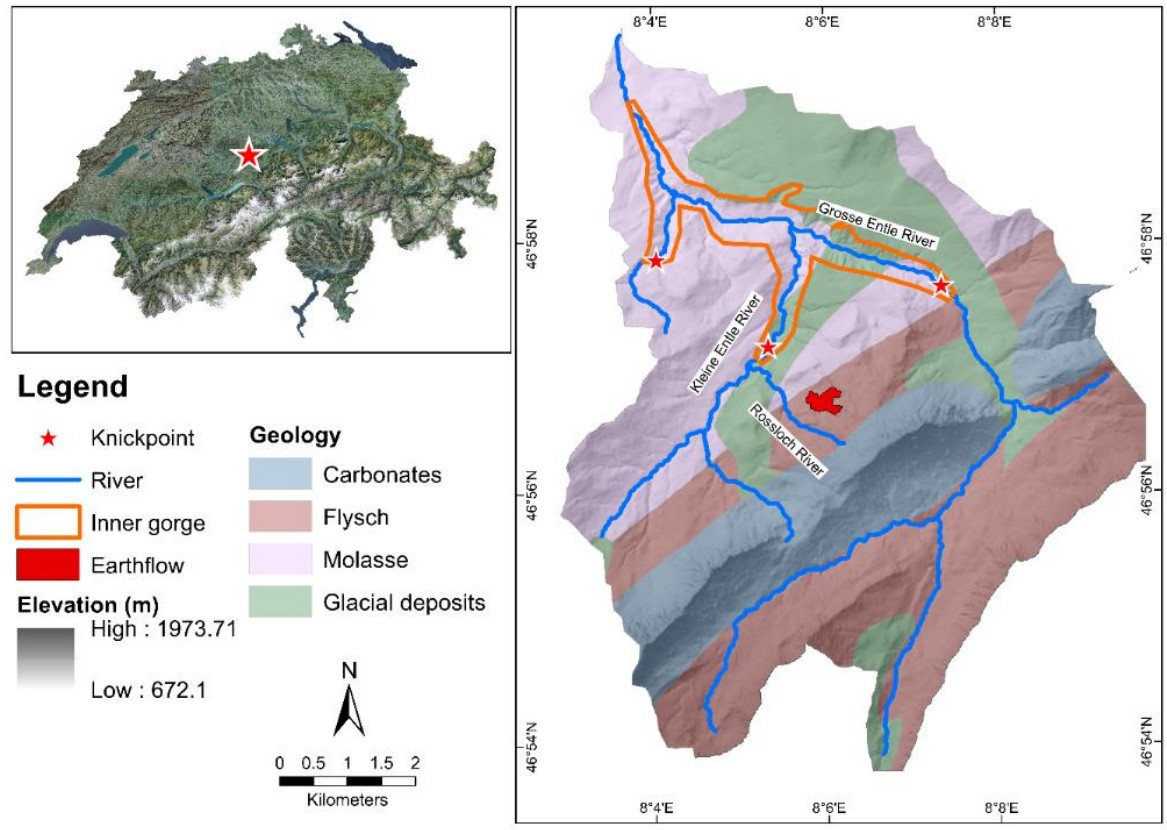

**Figure 1: Simplified geological and geomorphological settings of the Entle river catchment (after Van den Berg et al., 2012; Schlunegger et al., 2016b). Inset: Location of the study area in Switzerland.**

Within the Entle river catchment, the Schimbrig earthflow has particularly been active over the last 150 years (Clapuyt et al., 2017; Lopez-Saez et al., 2017; Savi et al., 2013; Schwab et al., 2008). It is located in the first-order Schimbrig catchment, the latter draining successively into the Rossloch River and the Kleine Entle, before entering the trunk river, i.e. the Grosse Entle. The earthflow occurs on the hillslopes of the Schimbrig ridge and is not physically connected to the Schimbrig stream, except during short episodes when superimposed debris flows occur. The Schimbrig earthflow consists of a fine-grained matrix of silt and mud, with centimetric to decimetric large clasts. The internal structure of the earthflow is complex with nested rotational units. Field observations revealed that a major earth slide with an up-to 12 m surface lowering occurred in the summer of 1994 after a succession of heavy precipitation events, followed by debris flows until March 1995 (Schwab et al., 2008). The intensity and spatial pattern of sediment redistribution, as well as the internal structure of the earthflow have rapidly evolved at the annual and decadal scale (Clapuyt et al., 2017).



## 2.2 Annual sediment fluxes

The annual sediment fluxes of the active part of the Schimbrig earthflow were derived from time series of very high-resolution topographic datasets from Clapuyt et al. (2017). The 3D topographic reconstructions were computed using the structure-from-motion algorithm (SfM) based on aerial photographs acquired by an unmanned aerial vehicle (UAV). Point clouds were

subsequently interpolated into digital surface models (DSM) at a spatial resolution of 0.04 m, defined based on the density of the point clouds. The accuracy of the time series is ranging between 0.20 m and 0.24 m. Acquisition dates are October 2013, June 2014 and October 2015. As part of this sediment flux assessment, only the dataset of the 2014-2015 time interval was used because it spatially covers the entire Schimbrig earthflow. The sediment budget was quantified by a digital elevation model of differences (DoD) between DSMs, using the Geomorphic Change Detection software (Wheaton et al., 2010). It

spatially assesses surface *lowering*, i.e. the decrease in ground elevation, and *bulging*, i.e. the increase in ground elevation, within the earthflow. Subsequently, the net difference between the volume of landslide deposits, i.e. *bulging* and eroded material, i.e. *lowering*, gives a first estimate of the net flux of landslide-derived material that is evacuated between June 2014 and October 2015. In this paper, values are eventually reported on an annual basis instead of over the entire period of interest (as in Clapuyt et al., 2017). Detailed information about the methodology and extended results are available in Clapuyt et al.

(2017). Errors reported at the annual scale were computed based on a uniform limit-of-detection applied on each topographic surface. Therefore, the associated error to the annual sediment fluxes should be seen as a maximum value. It is likely that spatializing the error on very high-resolution topographic measurements would lead to an increase of the signal-to-noise ratio.

## 2.3 Decadal sediment fluxes

The sediment fluxes at decadal scale were derived from Schwab et al. (2008), who assessed sediment yields of the Schimbrig

earthflow and associated slopes, i.e. the *Schimbrig catchment*, and linked it with sediment loads in the trunk stream, i.e. the *Waldemme River*. These authors based their analysis on a time series of DEMs derived from classic photogrammetry of aerial photographs acquired in 1962, 1986, 1993 and 1998. The associated volumetric errors of the photogrammetric workflow ranged between 2% for the 1986-1993 time interval and 29% for the 1962-1986 time interval. For this study, we converted the average sediment fluxes that were published in tons per year into cubic metre per year, using a material density of 2.70 g cm$^{-}$

$^3$ following a study by Gong (2005) on similar flysch units in Switzerland. Detailed methodology and results are available in the original paper of Schwab et al. (2008).

## 2.4 Millenial sediment fluxes

The geomorphic process rates at millennial scale were assessed from catchment-averaged denudation rates derived from in situ produced cosmogenic radionuclides in river-borne sediments. In order to get a comprehensive dataset on the spatial

variation in denudation rates in the Entle catchment, we collected river sand on eight locations in and around the Schimbrig catchment (Figure 2) and combined the resulting dataset with earlier work by Van den Berg et al. (2010). The new samples



were processed in a similar way as the sample preparation and processing described in Vanacker et al. (2007) and are similar to Van den Berg et al. (2010). After washing and sieving to the 0.25-1.00 mm fraction size, grains were separated using a Frantz Isodynamic Magnetic Separator. The remaining non-magnetic fraction was leached several times with 10% hydrochloric acid to remove organic, calcium and carbonate components. Then, samples were treated up to four times with 5% hydrofluoric acid in order to dissolve anything but quartz and also remove any meteoric $^{10}$Be left. After the leaching step, 157.8 µg of $^{9}$Be carrier was added to the clean quartz samples containing ca. 25 g of material. The purified quartz was subsequently dissolved in concentrated HF, from which beryllium is stepwise extracted using anion/cation exchange column chemistry. Remaining precipitates were oxidized and pressed into copper targets. Finally, $^{10}$Be/$^{9}$Be ratios were quantified using the 500 kV Tandy facility at ETH Zürich (Christl et al., 2013). These values were normalized with the in-house standard S2007N and corrected with a blank $^{10}$Be/$^{9}$Be ratio of $4.06 \pm 0.23 \times 10^{-15}$. Catchment-wide denudation rates were then computed from the in situ produced $^{10}$Be concentrations, i.e. from this study and from earlier data published by Van den Berg et al. (2010), using the catchment-averaged denudation rates from cosmogenic nuclide (CAIRN) method (Mudd et al., 2016). This open source calculator uses the topography to weight the $^{10}$Be production rate and shielding. A 1 m digital terrain model (DTM) resampled to 30 m resolution was used to compute topographic shielding. Snow shielding was averaged for each catchment individually. Snow cover is estimated using an elevation-dependent mean annual snow cover database for Switzerland (Auer, 2003). Following Jonas et al. (2009), an empirical relationship is used to derive the snow water equivalent thickness (SWE, g cm$^{2}$).We kept the default parameters from Mudd et al. (2016) to run the CAIRN model, including the sea-level high-latitude production rate of 4.30 at g$^{-1}$ yr$^{-1}$ (based on Braucher et al., 2011). Long-term denudation rates obtained by the CAIRN calculator were converted into sediment fluxes (m$^{3}$ yr$^{-1}$), by multiplying them with the catchment area.

## 3 Results

### 3.1 Schimbrig earthflow sediment dynamics at the annual scale

When focusing on the unstable part of the Schimbrig earthflow, we obtained a net mass flux of $1,000 \pm 4,000$ m$^{3}$ yr$^{-1}$ for the 2014-2015 period (Table 1; Clapuyt et al., 2017). Although this value is slightly positive, the mass flux is not significantly different from zero. The sediment budget for the 2013-2014 time interval supports this finding. In the eroding sites, the average denudation was $0.8 \pm 0.2$ m yr$^{-1}$, and equivalent to the accumulation that was observed in the bulging areas (Table 1). For the Schimbrig earthflow, we reported a mean horizontal displacement of ca. 6.30 m yr$^{-1}$ in the downslope direction (Clapuyt et al., 2017). The UAV-SfM derived data suggest that the earthflow has been in a dynamic equilibrium over the 2014-2015 period, with landslide-derived material being temporarily stored on the slopes during the period of interest.



**Table 1: Sediment fluxes for the 2014-2015 time interval from the Schimbrig earthflow (Modified from Clapuyt et al. (2017), to provide sediment fluxes reported on an annual basis).**

|  | Estimate | Error (±) |
|---|---|---|
| Total area of surface bulging (m² yr⁻¹) | 10763 | - |
| Total area of surface lowering (m² yr⁻¹) | 9730 | - |
| Average depth of surface bulging (m yr⁻¹) | 0.8 | 0.2 |
| Average depth of surface lowering (m yr⁻¹) | 0.8 | 0.2 |
| Average net depth of difference (m yr⁻¹) | 0.05 | 0.15 |
| Sediment flux (m³ yr⁻¹) | 1,000 | 4,000 |

Notwithstanding the state of dynamic equilibrium, our data suggest large internal movements with a complex pattern of sediment redistribution along the slope (Clapuyt et al., 2017). A succession of areas with terrain lowering and bulging characterised the earthflow along its longitudinal axis. The earthflow was re-adjusting to a new state of equilibrium after a massive failure that occurred in 1994. The sediment redistribution on the slopes was not associated with an increased sediment export downstream. Between 2013 and 2015, the hillslope domain was disconnected from the fluvial domain.

**3.2 Sediment budget of the Schimbrig catchment at decadal scale**

Sediment fluxes were computed from 1962 to 1998 over the first-order catchment affected by the earthflow, down to the confluence with the Rossloch River (Table 2; Schwab et al., 2008). The average sediment flux per year, evacuated from the catchment, is varying over time, from $14,000 \pm 4,000$ m³ yr⁻¹ for the 1962-1986 period, to $850 \pm 20$ m³ yr⁻¹ for 1986-1993 to $24,000 \pm 4,000$ m³ yr⁻¹ for the 1993-1998 time interval. The data from Schwab et al. (2008) indicate that there is no clear link

between the sediment fluxes exported from the Schimbrig catchment and the landslide dynamics on the hillslopes. This is evident from the fact that the proportion of the mass evacuated from the study area to the total displaced mass varies largely between 6% and 89% (Table 2).

**Table 2: Sediment fluxes between 1962 and 1998 (after Schwab et al., 2008).**

| Time interval | Average sediment flux evacuated (m³ yr⁻¹) | Error Sediment flux (m³ yr⁻¹) | Percentage of mass evacuated compared to the displaced mass (%) |
|---|---|---|---|
| 1962-1986 | 14,000 | 4,000 | 89 |
| 1986-1993 | 850 | 20 | 6 |
| 1993-1998 | 24,000 | 4,000 | 34 |
| 1962-1998 | 13,000 | 2,000 | 78 |

During the 1962-1986 period, about 89% of the displaced mass was evacuated from the catchment, suggesting that sediment storage during this period was not significant. In contrast, during the following periods (1989-1993, 1993-1998) when the major earthflow event occurred in 1994, only 34% of the landslide material was evacuated.



### 3.3 Long-term denudation rates of the Entle catchment

Overall, the CRN-derived denudation rates (Table 3; Figure 2; Van den Berg et al., 2010) range from $160 \pm 30$ mm kyr$^{-1}$ in the upper part of the Entle catchment, i.e. the Rothbach River (*E-9*), and $5000 \pm 1000$ mm kyr$^{-1}$ in the upper part of the Schimbrig earthflow (*CH-ENT-3*). Importantly, measurements from Van den Berg et al. (2010) and from this study are spatially

and quantitatively consistent with each other (Figure 2). As the $^{10}$Be concentrations of the landslide-affected area were low, relatively high errors were reported for the $^{10}$Be concentrations of the landslide-derived sediment material (Table 3). The errors subsequently propagated on the denudation rates, particularly for the samples from the Schimbrig stream draining the earthflow, i.e. *CH-ENT-1*, *CH-ENT-5* and *CH-ENT-8*.

Our data show two opposite patterns of denudation rates and sediment fluxes in relation to drainage area and downstream

distance, which are driven by the landslide occurrence (Table 3; Figure 3; Figure 4).

**Table 3: Denudation rates (mm kyr$^{-1}$) and sediment fluxes (m$^3$ yr$^{-1}$) for the Schimbrig catchment and other first-order rivers, i.e. samples CH-ENT-*, from this study and re-computed denudation rates and sediment fluxes for the Entle River catchment, i.e. samples Ent* and E-*, from Van den Berg et al. (2010).**

| Sample | Drainage area (km$^2$) | Altitude (m) | Latitude (dd) | Longitude (dd) | $^{10}$Be concentration ($\times 10^3$ at g$^{-1}$) | Production scaling | Topographic shielding | Snow shielding | Denudation rate (mm kyr$^{-1}$) | Sediment flux (m$^3$ yr$^{-1}$) |
|---|---|---|---|---|---|---|---|---|---|---|
| CH-ENT-1 | 0.20 | 1451 | 46.945 | 8.098 | $0.62 \pm 0.45$ | 3.306 | 0.954 | 0.904 | $1,000 \pm 2,000$ | $300 \pm 400$ |
| CH-ENT-2 | 1.96 | 1264 | 46.951 | 8.086 | $2.94 \pm 0.28$ | 2.965 | 0.974 | 0.917 | $260 \pm 60$ | $500 \pm 100$ |
| CH-ENT-3 | 0.16 | 1500 | 46.943 | 8.104 | $0.15 \pm 0.53$ | 3.427 | 0.937 | 0.900 | $5,000 \pm 1,000$ | $900 \pm 200$ |
| CH-ENT-5 | 0.35 | 1373 | 46.946 | 8.097 | $2.32 \pm 0.58$ | 3.208 | 0.961 | 0.909 | $300 \pm 100$ | $120 \pm 40$ |
| CH-ENT-6 | 3.25 | 1306 | 46.942 | 8.077 | $4.12 \pm 0.28$ | 3.069 | 0.950 | 0.914 | $180 \pm 40$ | $600 \pm 100$ |
| CH-ENT-7 | 4.54 | 1336 | 46.942 | 8.076 | $2.57 \pm 0.27$ | 3.112 | 0.960 | 0.912 | $300 \pm 70$ | $1,400 \pm 300$ |
| CH-ENT-8 | 0.51 | 1325 | 46.947 | 8.092 | $1.34 \pm 0.27$ | 3.033 | 0.970 | 0.913 | $600 \pm 200$ | $290 \pm 80$ |
| CH-ENT-9 | 1.07 | 1287 | 46.946 | 8.092 | $3.53 \pm 0.28$ | 2.983 | 0.977 | 0.915 | $210 \pm 50$ | $230 \pm 50$ |
| Ent3-1 | 3.32 | 1076 | 46.9561 | 8.0621 | $1.94 \pm 0.11$ | 2.434 | 0.979 | 0.954 | $340 \pm 70$ | $1,100 \pm 200$ |
| Ent4-1 | 54.6 | 1323 | 46.9367 | 8.1049 | $2.43 \pm 0.12$ | 3.007 | 0.965 | 0.948 | $320 \pm 70$ | $17,000 \pm 4,000$ |
| E-5 | 26.53 | 1482 | 46.9235 | 8.1157 | $3.11 \pm 0.17$ | 3.357 | 0.956 | 0.937 | $270 \pm 60$ | $7,000 \pm 1,000$ |
| E-7a | 63.56 | 1274 | 46.9411 | 8.1 | $2.46 \pm 0.13$ | 2.922 | 0.967 | 0.951 | $310 \pm 60$ | $20,000 \pm 4,000$ |
| E-8 | 8.05 | 1547 | 46.9073 | 8.1074 | $4.60 \pm 0.24$ | 3.635 | 0.959 | 0.933 | $200 \pm 40$ | $1,600 \pm 300$ |
| E-9 | 7.83 | 1537 | 46.9159 | 8.0881 | $5.28 \pm 0.26$ | 3.369 | 0.967 | 0.933 | $160 \pm 30$ | $1,300 \pm 300$ |
| E-10 | 3.13 | 1445 | 46.9417 | 8.1527 | $4.70 \pm 0.21$ | 3.326 | 0.975 | 0.935 | $180 \pm 40$ | $600 \pm 100$ |
| E-11 | 16.01 | 1231 | 46.9393 | 8.0814 | $2.27 \pm 0.13$ | 2.871 | 0.969 | 0.949 | $330 \pm 70$ | $5,000 \pm 1,000$ |
| E-12 | 11.71 | 1280 | 46.934 | 8.0763 | $3.53 \pm 0.25$ | 2.989 | 0.964 | 0.946 | $220 \pm 50$ | $2,500 \pm 500$ |
| E-13 | 16.64 | 1534 | 46.9125 | 8.0988 | $5.22 \pm 0.24$ | 3.511 | 0.960 | 0.934 | $170 \pm 30$ | $2,800 \pm 600$ |

Ignoring samples from landslide-affected catchments, long-term denudation rates correlate positively with downstream distance, with values ranging from $160 \pm 30$ mm kyr$^{-1}$ (*E-9*) to $340 \pm 70$ mm kyr$^{-1}$ (*Ent3-1*). Characterised by a low variability, these values are similar to other denudation rates found in the similar alpine tectonic settings (e.g. Norton et al., 2008).



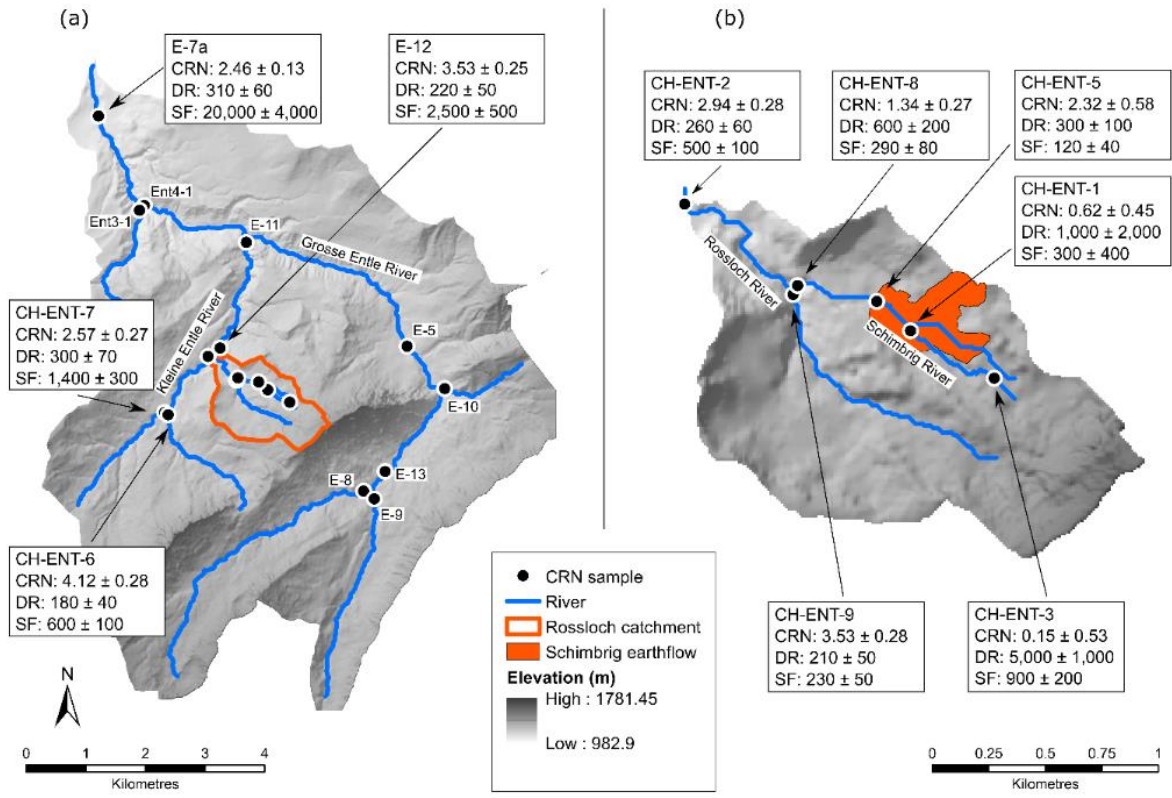

**Figure 2: Location of CRN samples in the Entle catchment. CRN concentrations (CRN; ×10³ at g⁻¹), denudation rates (DR; mm kyr⁻¹) and sediment fluxes (SF; m³ yr⁻¹) values displayed are discussed in the text. Other values are available in (Table 3). (a) Entle River and (b) Rossloch River catchments.**

5    This increase in denudation rates with downstream distance is triggered by the ongoing relief rejuvenation and incision of the inner gorge after the LGM (Van den Berg et al., 2012). The data also indicate that the river network effectively evacuates sediments that are supplied to the river channel. Accounting for the drainage area at each sample location, long-term sediment fluxes (Table 3; Figure 4) show the same positive correlation with downstream distance as denudation rates.





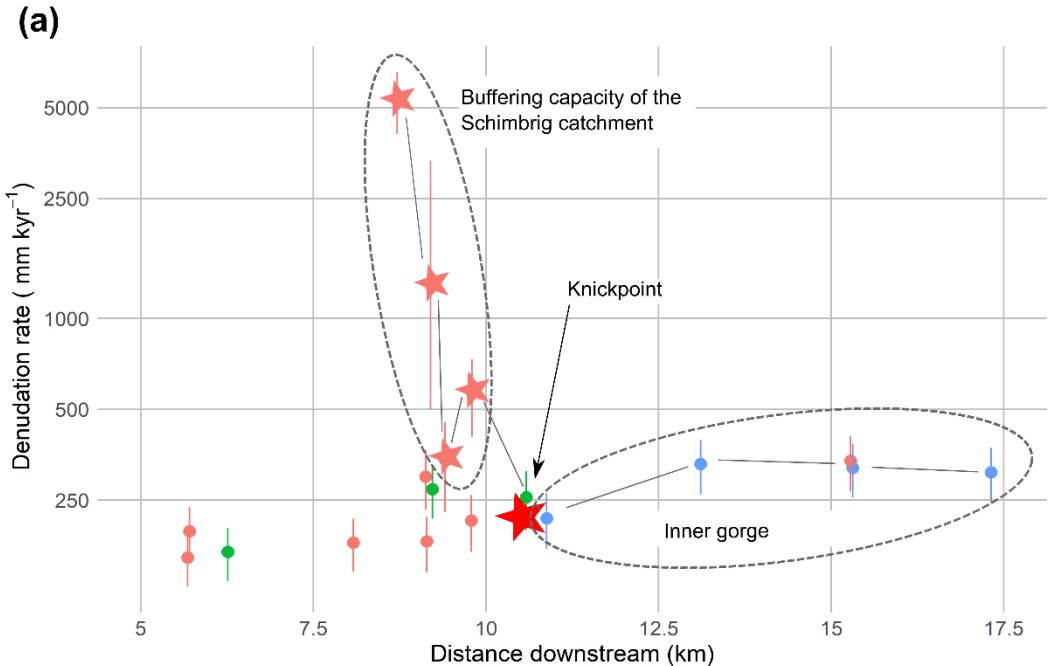

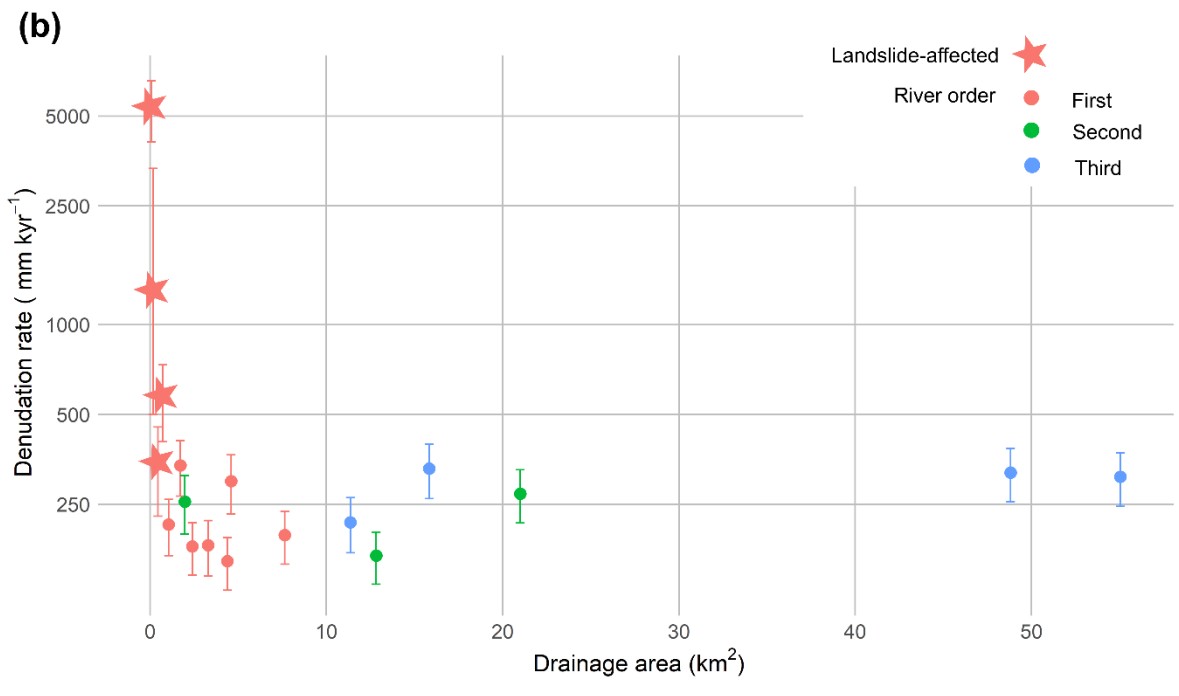

**Figure 3: Denudation rates (mm kyr⁻¹) in the Entle catchment against (a) downstream distance and (b) drainage area. Downstream distance is computed from river source to outlet of the Entle catchment. Marker *star* represents CRN-derived denudation rate from the landslide-affected Schimbrig catchment. Grey line presents the downstream sequence of denudation rates along the Schimbrig, Rossloch, Kleine Entle and Grosse Entle rivers.**





The sediment fluxes range from $600 \pm 100$ m³ yr⁻¹ in upper first-order catchments (*CH-ENT-6* and *E-10*) to $20,000 \pm 4,000$ mm kyr⁻¹ at the outlet of the Grosse Entle River (*E-7a*). This two-order-of-magnitude increase in sediment fluxes downstream corroborates the inferred efficiency of sediment evacuation by the fluvial system.

The landslide-affected catchments show a different pattern that deviates from the overall trend of increasing denudation rates with distance downstream and catchment area. In the first-order Schimbrig catchment, long-term denudation rates correlate negatively with downstream distance (Table 3; Figure 2; Figure 3).

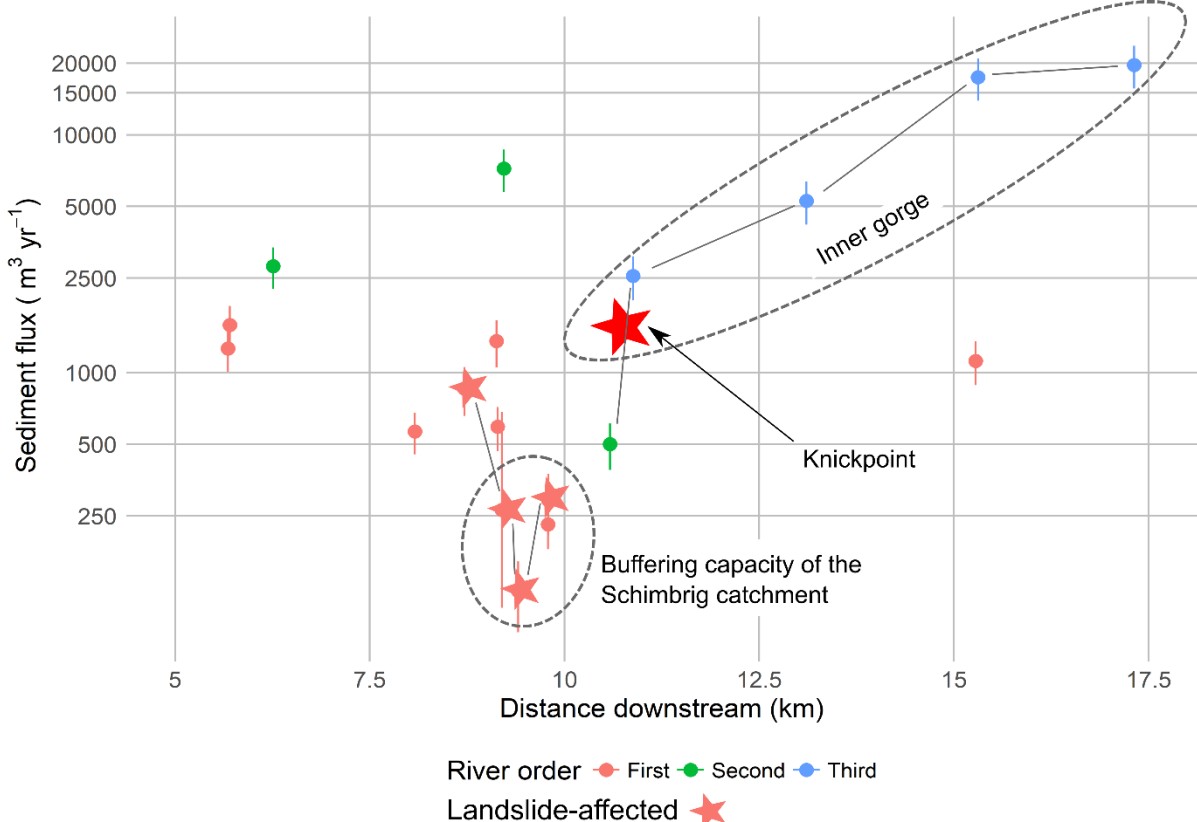

**Figure 4: Sediment fluxes (m³ yr⁻¹) along the river network of the Entle catchment. Downstream distance is computed from river source to outlet of the Entle catchment. Marker *star* represents sediment flux from the landslide-affected Schimbrig catchment. Grey line presents the downstream sequence of denudation rates along the Schimbrig, Rossloch, Kleine Entle and Grosse Entle rivers.**

The denudation rates of the landslide-affected first-order catchment are up-to one order of magnitude higher than the rest of the Entle catchment, with values ranging between $300 \pm 100$ mm kyr⁻¹ in the intermediate part (*CH-ENT-5*) and $5,000 \pm 1,000$ mm kyr⁻¹ (*CH-ENT-3*) in the upper part of the Schimbrig catchment. The variability in denudation rates along the river is very high, with a one-order of magnitude difference between minimum and maximum values, and spatially highly variable. For example, the denudation rate in *CH-ENT-5* is lower than the sites directly up- and down-stream of this sampling location, reflecting the stochastic character of the sediment delivery from the earthflow. The decrease in denudation rates with catchment



area suggests that material initially displaced by the earthflow is not directly evacuated to the river network but remains on the slopes, hence accumulating $^{10}$Be atoms in sediments.

Given that the catchment area of the first-order landslide-affected catchments is small, the absolute sediment fluxes are low with values ranging between $120 \pm 40$ m$^3$ yr$^{-1}$ and $900 \pm 200$ m$^3$ yr$^{-1}$. These long-term sediment fluxes computed for the Rossloch catchment, i.e. including the Schimbrig area, are one to two orders of magnitude lower than values computed in other parts of the Entle catchment (Figure 4).

The impact of landsliding on the long-term sediment dynamics can be evaluated by comparing the denudation rates of the two intersecting catchments at the confluence with the Rossloch river. At the outlet of the Schimbrig catchment, the denudation rate, i.e. $600 \pm 200$ mm kyr$^{-1}$ (*CH-ENT-8*), is at least two times higher than in the neighbouring catchment, i.e. $210 \pm 50$ mm kyr$^{-1}$ (*CH-ENT-9*). When accounting for their catchment area, the sediment fluxes are very similar with values of respectively $290 \pm 80$ m$^3$ yr$^{-1}$ (*CH-ENT-8*) and $230 \pm 50$ m$^3$ yr$^{-1}$ (*CH-ENT-9*).

## 4 Discussion

### 4.1 Temporal upscaling: The stochastic nature of landsliding

Characterised by a relatively gentle alpine topography, the millennial geomorphic activity of the study area is moderate in intensity. In comparison, the mean CRN-derived denudation rate is $270 \pm 140$ mm kyr$^{-1}$ in similar surrounding areas, i.e. in the Alpine foreland, but increases to $900 \pm 300$ mm kyr$^{-1}$ in the high crystalline Alps (Wittmann et al., 2007). The Entle River catchment was affected by an active earthflow at least over the past 150 years (Lopez-Saez et al., 2017; Savi et al., 2013). Given the extent of its active part, i.e. ca. 0.5 km$^2$, the Schimbrig earthflow is a larger-than-average landslide (Stark and Hovius, 2001). Therefore, according to the magnitude-frequency distribution of landslides (e.g. Hovius et al., 1997), this type of occurrence is generally relatively infrequent, i.e. between $10^{-2}$ km$^{-2}$ yr$^{-1}$ and $10^{-3}$ km$^{-2}$ yr$^{-1}$. The mean annual horizontal displacements that we measured within the earthflow, i.e. ca. 6.30 m yr$^{-1}$ for the period 2014-2015, are high in comparison with the decimetric displacements reported in the Western Slovakian Carpathians (Prokešová et al., 2014), $< 2$ m yr$^{-1}$ for the Eel earthflow in California (Mackey et al., 2009), and metric to decametric displacements for the Super-Sauze landslide in Southern French Alps (Niethammer et al., 2012).

At the annual scale, the Schimbrig earthflow experienced intense sediment redistribution, with a mean horizontal surface displacement of more than 6 m yr$^{-1}$. The sediment budget of the Schimbrig earthflow is roughly in equilibrium, with the occurrence of very limited sediment evacuation from the hillslope domain to the fluvial domain. This implies that the Schimbrig earthflow was not coupled to the channel network for the period 2013-2015. These results support the idea that landslide processes act as stochastic sediment pulses, producing sediments on hillslopes in the landscape. Their impact is attenuated as follows over short time scales: (1) After the principal slope failure, only a small fraction of the mobilised sediment volume is exported to the river network. (2) Subsequently, the internal sediment reorganisation on the hillslopes results in very dynamic topography with very limited sediment evacuation to the channels. Hence, at the decadal scale, sediment fluxes





computed over the entire Schimbrig catchment support the episodic pattern of connection/disconnection between hillslopes and channels. Sediment transfer of landslide colluvial fans to the river network only occurs by superimposed stochastic debris flows, the latter being the main mechanisms of hillslope-channel coupling. Schwab et al. (2008) suggested that this pattern occurs at the decadal scale. They showed that an intense geomorphic activity within the Schimbrig catchment is not associated

with phases of increased sediment export to the channel network, and resulted in storage of landslide-derived debris on the hillslopes even during extended or intense rainfall episodes. For instance, the large slope failure of 1994, which lowered the ground surface up to 12 m, produced fresh sediments that were temporarily stored as colluvial material on the hillslopes instead of being directly transported to the river network. In contrast, phases of lower hillslope geomorphic activity were coinciding with higher sediment export to the river channel and evacuation of landslide-debris from the hillslope to the fluvial domain.

During these periods, mobilised sediments by previous landslides are gradually and/or episodically transported downstream.

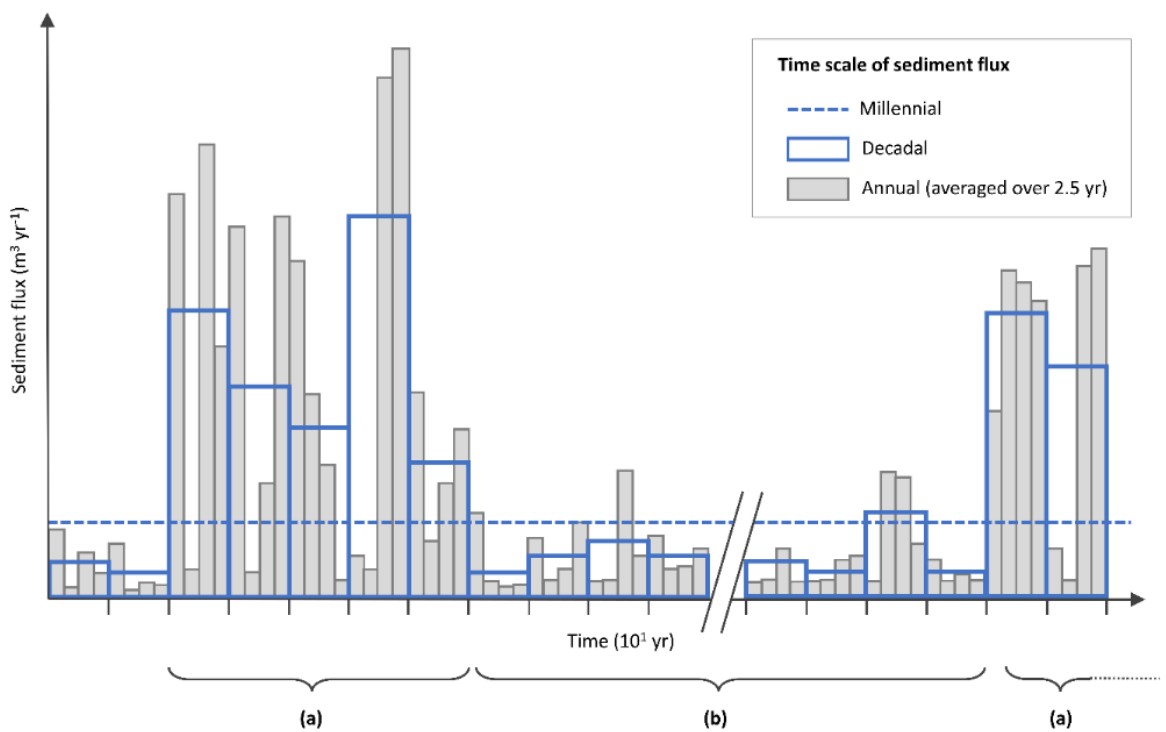

**Figure 5: Conceptual representation of the sediment fluxes over different time scales at the outlet of a first-order river catchment. (a) Periods of high hillslope activity due to a landslide sediment pulse, resulting in intermittent hillslope-channel connections and in sediment export to the river network at the annual scale. (b) Periods of low hillslope activity both at the annual and decadal scales,**
**due to poor sediment availability on slopes.**

At the millennial scale, catchment-wide denudation rates show a positive relationship with downstream distance and drainage area, if landslide-affected catchments are ignored. The latter are characterised by a negative relationship with downstream distance, along with a high variability. Interestingly, the large difference in denudation rates between the Schimbrig (*CH-ENT-8*) and upper Rossloch (*CH-ENT-9*) catchments indicates that the sediment dynamics of both catchments were very different



over the last thousands of years. The high variability of denudation rates in headwaters (Figure 3a) emphasizes the episodic character of landslides. When accounting for the surface area of the catchments, the difference between the two adjacent streams disappears. As such, the landslide-affected Schimbrig catchment only has a small impact on the overall sediment budget of the Entle River catchment.

In addition to the hillslope-channel connectivity, our database of sediment fluxes also highlights the non-steady character of geomorphic processes. The average decadal sediment flux of the Schimbrig catchment, i.e. $13,000 \pm 2,000$ m$^3$ yr$^{-1}$ between 1962 and 1998 (Table 2), is in fact two orders of magnitude higher than sediment fluxes computed at the millennial time scale over the same spatial extent, i.e. $290 \pm 80$ m$^3$ yr$^{-1}$ (*CH-ENT-8*). In other words, the decadal sediment flux is not representative for the long-term, millennial sediment flux of the first-order river catchments (Figure 5).

**4.2 Spatial upscaling: The importance landsliding for sediment budgets**

Taking benefit of our multi-temporal database, the mismatch between annual, decadal and millennial sediment fluxes is probably caused by the episodic character of sediment fluxes over long time scales. Since the oldest photogrammetric dataset, i.e. 1962, the Schimbrig catchment experienced active landsliding. Dendrogeomorphological data confirmed that slope movements were taking place over the last > 150 years (Lopez-Saez et al., 2017; Savi et al., 2013). However, all other

parameters remaining equal, our data suggest that this situation is temporary, and was preceded by a long phase of inactivity. The transient character of the important geomorphic activity in the Schimbrig catchment, i.e. a first-order river, is eventually supported by following observation: its decadal sediment flux, i.e. $13,000 \pm 2,000$ m$^3$ yr$^{-1}$ (Table 2), is of the same order of magnitude than the background sediment flux going out of the entire Entle River catchment, i.e. $20,000 \pm 4,000$ m$^3$ yr$^{-1}$ (*E-7a*; Table 3; Figure 2). In other words, it means that over the last 50 years, the Schimbrig catchment, which represents ca. 1% of

the entire study area, provides 65 % of the sediments that the entire Entle catchment will supply by over the millennial scale. Subsequently, sediment transport will attenuate the enormous sediment pulse from the Schimbrig to the outlet of the Entle catchment. Consequently, the impact of the Schimbrig earthflow is currently important with respect to the rest of the Entle catchment, after an initial major slope failure more than 150 years ago. The higher-order river network, i.e. the Kleine and Grosse Entle rivers, attenuates decadal sediment pulses by sediment transport, during which subsequent erosion and deposition

occur over longer time scales. In the future, the slopes of the Schimbrig will find a new equilibrium after the on-going landslide perturbation, leading to a lower geomorphic activity. The final state of equilibrium will be reached when the entire volume of sediments mobilised by the initial failure and its reactivations will be depleted from the hillslope. Somehow, due to the stochastic character of landslides, a new sediment pulse will affect another sub-catchment of the Entle area, inducing a new cycle. Accordingly, landslides affecting hillslopes over short time scales are in fact considered as noise over long time scales.

They trigger localised sediment pulses, which are subsequently shredded by sediment transport when reaching the channel network. This noise is eventually averaged out when considering longer time scales and larger spatial scales. Consequently, one single landslide has not necessarily an impact on the long-term sediment budget of first-order catchments. Rather, it is the



cumulated effect of multiple landslides, which are intermittently connected to the channel network at the decadal scale, along with sediment transport, that may regulate sediment fluxes at the regional scale over the millennial time scale.

### 4.3 Conceptual upscaling: refinement of the sediment connectivity concept for landsliding

At a broader scale, the results of our work allow to refine our conceptual understanding of how the sediment connectivity

5   conditions the routing of material from the source to the sink. We conceptualize this cascade of sediment transfer at the example of the Schimbrig landslide and the Entlen drainage basin (Figure 6). In such a framework, landslides, situated on hillslopes, act as sediment factories and initiate the mobilisation of regolith, leading subsequently to stochastic sediment pulses. After landslide initiation, an important fraction of the landslide-derived material is not directly transferred to the river network, and accumulates on the slopes as landslide colluvial fans when hillslopes are not coupled to the channel network.

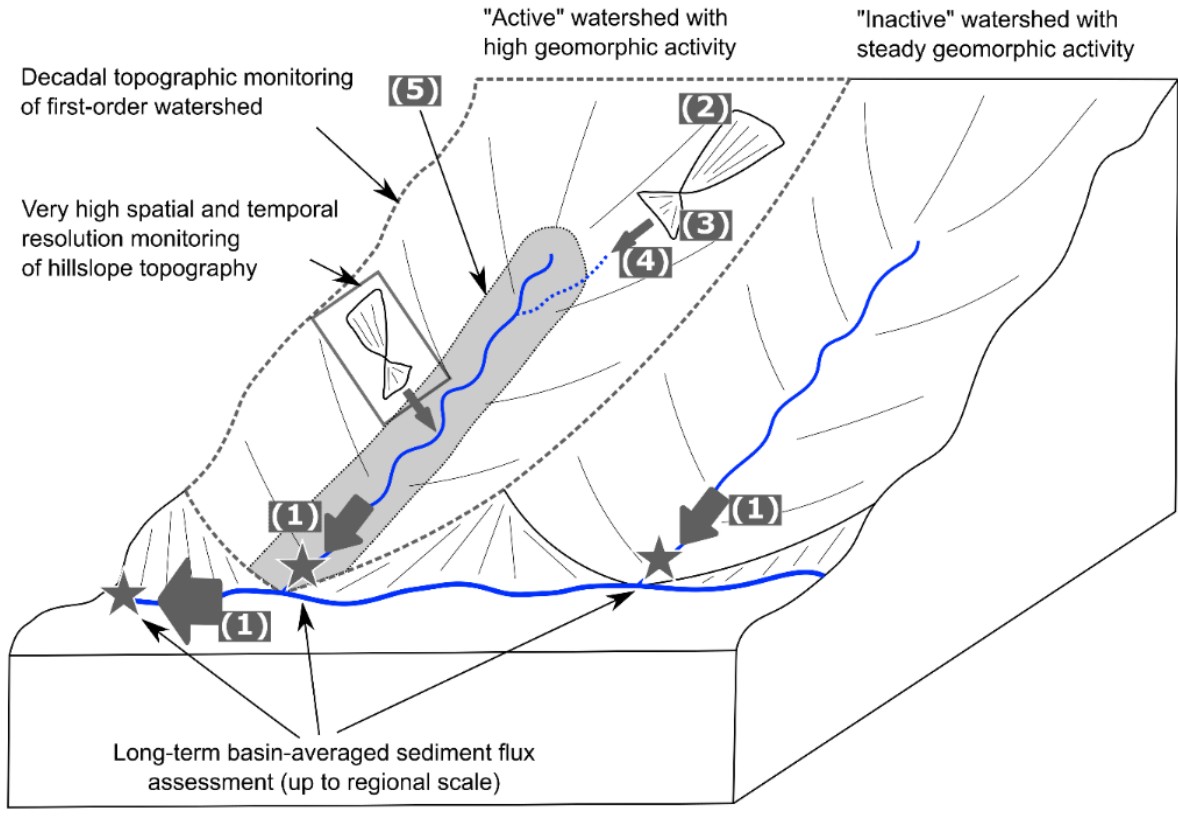

**Figure 6: Conceptual framework of the sediment dynamics in alpine landslide-affected first-order catchments. (1) Long-term sediment flux from first-order catchments. (2) Sediment production and mobilisation on hillslopes, i.e. stochastic sediment pulses driven by landslide processes. (3) Temporal sediment storage on hillslopes, i.e. landslide colluvial fans. (4) Hillslope-channel connectivity-driven sediment flux. (5) Zone of geomorphic coupling between adjacent hillslopes and channels, through e.g. slope**

15   **failures induced by river undercutting.**





The local accumulation of landslide deposits often goes along with an intense sediment reorganisation on the slopes to reach a new state of equilibrium (Clapuyt et al., 2017). Subsequently, superimposed debris flows, surface creep or gully channels control the transfer of sediment from colluvial deposits to the river network, during short periods of geomorphic coupling between hillslopes and channels (Schwab et al., 2008). These episodes of geomorphic coupling are often triggered by extreme

precipitation or seismic events. In upstream mountainous catchments, the fluvial network is often sediment supply-limited and the sediment is then effectively transported by the river to downstream reaches (Van den Berg et al., 2012).

Besides the spatial connectivity between hillslopes and channels described above, the following temporal aspects condition the propagation of a landslide-driven sediment pulse along the sediment cascade. When using direct observations, typically collected over several decades, only few high-magnitude low-frequency landslide occurrences can be observed. When active,

the landslides mobilise a large amount of regolith on the hillslopes and can have a significant impact on the sediment dynamics. When inactive, other geomorphic processes like e.g. water erosion and soil creep are contributing sediment to the river system, resulting in a lower than average sediment supply to the river network. The pattern and rate of episodic landsliding can be quantified at annual and decadal time scales using direct measurements, and their frequency will control the long-term sediment mobilisation in the hillslope domain. Because sediment transport acts as a nonlinear filter that destroys environmental signals

(Jerolmack and Paola, 2010), the signal of the landslide-driven sediment pulse is dampened along the sediment cascade. Hence, the high geomorphic process rates that we can observe in the landslide-affected hillslope domain are not necessarily transmitted to the river system. This concept thus illustrates how stochastic sediment pulses are dampened at a larger spatial and temporal scale.

Attempting to capture sediment pulse propagation along the sediment cascade inherently implies integrating different spatio-

temporal scales: from the annual and decadal to millennial time scale, and from the hillslope domain where landslides occur to the colluvial and fluvial domains where sediments are transported downstream. The time scale dependency of geomorphic processes is known as the Sadler effect. Initially suggested to highlight the completeness of stratigraphic records, it outlines that an inverse linear relationship exists between one-dimensional sedimentation rates and the time span of measurements, because of the unsteady and discontinuous character of sedimentation (Sadler, 1981). This time scale dependency has also

been associated with periods of stasis or inactivity which occur episodically over long time scales and decrease measured rates (Paola et al., 2018). When assessing sediment dynamics at the hillslope-channel domains, the time scale dependence which might compromises one-dimensional rates becomes less evident to assess (Sadler and Jerolmack, 2015). At the landscape scale, the catchment-wide rates of denudation or deposition can capture a range of geomorphic processes that all contribute in different proportions to the measured denudation rates and integrate periods of activity and inactivity. Moreover, Sadler and

Jerolmack (2015) showed that there is no compelling dependence between upland denudation rates and averaging time, except at time scales shorter than a month.



## 5 Conclusion

To better understand mechanisms and scales of geomorphic connectivity in mountainous environments, we characterised the sediment cascade of the Entle River catchment located in the foothills of the Central Swiss Alps. We quantified (or took benefit from previous studies of) sediment fluxes over annual, decadal and millennial time scales using respectively UAV-SfM-based

3D topographic reconstructions, classic photogrammetry and in situ [10]Be cosmogenic radionuclides.

Our database of sediment fluxes supported three hypotheses about the dynamics of the sediment cascade in first-order catchments and the dynamic feedback between hillslopes and channels: (1) In the landscape, landslides act as local sediment pulses, making them available for further mobilisation and transport when hillslopes and channels are connected. (2) Connection/disconnection cycles occur at the decadal scale. (3) High decadal geomorphic activity at the catchment scale is

episodic over the millennial temporal scale. At the annual scale (2013-2015), the mass balance of the Schimbrig earthflow is in equilibrium despite the intense sediment redistribution on hillslopes, which indicates that the latter is disconnected from the channel network. Hillslope-channel connection occurs at the decadal scale (1962-1998) during stochastic rainfall-triggered superimposed debris flows, causing sediment supply to the river network. At the millennial scale, higher-than-average long-term denudation rates characterise the intense geomorphic activity of the Schimbrig catchment. However, the high-magnitude

low-frequency character of the Schimbrig earthflow implies that it does not affect the long-term sediment flux pattern. In addition, nothwithstanding the intense sediment activity of the Schimbrig earthflow at the decadal scale, it only affects a small first-order river catchment.

Our data highlight the episodic character of sediment fluxes from first-order river catchments in alpine environments and the attenuation of sediment pulses by sediment transport throughout landscapes. Landslides act as stochastic but discrete sediment

sources in the landscape and randomly affect these catchments. During short periods occurring at the decadal scale, when hillslopes geomorphically connect to channels, the sediment flux of the Schimbrig catchment is of the same order of magnitude than the total sediment flux of the Entle catchment. This implies that during these short phases of connection, more than 65 % of the sediment is coming from 1 % of the total catchment area. However, during 90 % of time, hillslope-channel system of the first-order catchment is uncoupled from the lower reaches at the regional scale.

When the sediment source will be depleted, another first-order river catchment will be affected by landslides and will provide enough sediments to match the long-term regional sediment flux. Consequently, one single landslide has not necessarily an impact on the long-term sediment budget of mountainous river catchments at the regional scale. It is the integrated effect of multiple sediment pulses providing sediments to the river network at the decadal scale, which are subsequently attenuated by sediment transport that may regulate sediment fluxes at the regional scale.




*Data availability.* All datasets and code are available upon request. Please contact the first author for details.

*Competing interests.* The authors declare that they have no conflict of interest.

*Acknowledgments.* The Authors would like to thank Jérome Schoonejans for the extended help and the careful supervision of the entire CRN extraction procedure in the Cosmo Laboratory of the Georges Lemaître Centre for Earth and Climate Research at the Université catholique de Louvain. We extend our thanks to Marco Bravin for the help with sample leaching in the soil lab. We also acknowledge Romain Delunel from Bern Universität for the computation of the snow shielding factors which
10 serves as inputs to derive long-term denudation rates. Finally, efficient and reproducible research is possible thanks to open-source algorithms, such as CAIRN, i.e. the CRN-derived denudation rates calculator, developed by Simon Mudd and his colleagues.



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
