# Peer review of "Spatio-temporal dynamics of sediment transfer systems in landslideprone alpine catchments"

_Solid Earth, 2018_

## Referee Comment (RC1) · Luca C Malatesta (Referee) · 20 Feb 2019

Dear editor,

Clapuyt and colleagues present new constraints on geomorphic activity in an alpine catchment from cosmogenic radionuclides and from drone surveys (structure for motion). This new data complements existing constraints on sediment fluxes and allows a view across timescales from $10^0$ to $10^3$ yrs. The authors observe that the episodic activity of one earthflow leads to the production of a volume of sediment equivalent to that of the entire catchment, but that the evacuation of this material is limited by transport efficiency through the catchment such that earthflow activity is unlikely to imprint

the stratigraphic record.

The science in this paper seems sound. I simply have one concern regarding potential recycling of glacial sediment for the cosmogenic radionuclide erosion rates. And I am confused by the interpretation of the catchment as a supply-limited vs. transport-limited system.

The language and the figures of the article are good. Though the dynamics of the earthflow are too summarily described and I have not really understood what is actually measured.

The manuscript, however, could be much improved by reworking its structure. At the moment, the novel contribution of the authors is somewhat buried under a discussion of known elements. I strongly encourage the authors to rethink the introduction and the motivation of their study to increase its impact. I provide some suggestions below.

All in all I recommend to accept this manuscript once the issue of potential sediment recycling is addressed and after 1) the sediment dynamics of the earth flow are more clearly defined and 2) the novel elements of the article are better highlighted.

Sediment recycling: Figures 3 and 4 show a downstream increase in erosion rates and sediment fluxes once the Entle river flows in the inner gorge. The authors attribute this to the fast rate of postglacial incision in the gorge. To me, it however seems that the recycling of buried glacial sediment could be at least in part responsible for the trend of apparent increase in erosion rates caused by the increasing admixture of sediment with lower CRN concentration. If that is not a driver behind the increase in incision rate it should be explained. And if, on the contrary, this plays a role, this should be quantified.

Meaningfulness of the earthflow sediment dynamics: I do not think that I correctly understood what was being surveyed and what that entails for the sediment cascade. I have commented several parts of the manuscript (attached) where I might have been
confused by a lack of clarity. Surface lowering on the earthflow is described as being the result of erosion (p. 6 l. 10-13). Isn't it also due to subsidence of the surface? I would expect erosion to mainly affect the bulging parts of the flow and mitigate the rate of surface uplift. Are subsidence and erosion differentiated? The lowering and rising parts of the flow both do so at the exact same mean rate. This is a rather arresting coincidence. It could be useful to add one sentence to explain/confirm this to avoid it being perceived as a red flag! The net mass flux of the earthflow is close to zero. This implies a constant volume though time. Then wouldn't the throughput flow, instead of the net balance, be the quantity that matters for sediment yield from the earthflow? Or alternatively, considering firstly the flux from bedrock to sediment (production) and secondly the loss of sediment from the earthflow to the channel (transfer). It would be potentially useful to reproduce a figure of deformation (bulging/lowering) on the earthflow in the manuscript to contextualize the values.

Sediment system: transport-limited or supply-limited? The catchment is framed as being supply-limited (p. 4 l. 24 and p. 17 l. 6-7). But it seems that the authors provide arguments for it being transport-limited — at least in the first orders tributaries (p. 15 l. 25) — the two conflicting accounts need to be reconciled. It is possible that a supply-limited catchment switches to being transport-limited when a landslide pulse overwhelms the transport paths.

Structure: As it stands, I find that the article fails to properly motivate the study and to highlight the novelty the authors provide. In the discussion section, the authors use a significant amount of space to present already well-established conceptual models (sediment cascades, buffering of sediment pulses, stochasticity of landslides). The effect is to dilute the author's work. I believe it would be much more effective to introduce all these known/established elements at the beginning of the manuscript. This would allow the authors to explicitly define the gap in knowledge that their work directly addresses: a dataset across timescales, and not a conceptualisation of sediment transfer. I believe this would make it easier for the reader and increase the impact of the

presented work. This section would also be the good place where to describe how the different processes affecting earthflow dynamics contribute to the sediment routing system.

I would like to encourage the authors to make better use of their data. Instead of synthetic data on the last figure, why don't they actually plot a distribution of erosion rates vs. timescale of integration (not time!) to present what is their truly significant contribution (data across timescales)? See Fig. 1 of Sadler 1981 for reference.

The authors will find my line-by-line comments in the annotated pdf file hereby.

If any of my comments are unclear, the authors are welcome to contact me for clarification.

Kind regards, Luca Malatesta

Please also note the supplement to this comment:
https://www.solid-earth-discuss.net/se-2018-139/se-2018-139-RC1-supplement.pdf

[Figure]

**Supplement:**

[revised manuscript text omitted]

---

## Referee Comment (RC2) · Anonymous Referee #2 · 26 Feb 2019

Dear Editor,

Clapuyt et al. investigate the sediment contribution of the Schimbrig earthflow (Switzerland) to the sediment flux of the entire Entle catchment with a particular focus of hillslope-channel coupling. They quantify the sediment flux of the earthflow on annual, decadal and millennial timescales by combining previously published data of sfm-analysis of aerial photographs (annual) and time-series of photogrammetry-derived DEMs (decadal) with new and previously published 10Be-derived denudation rates (millennial). They conclude that sediment contribution from the earthflow to the fluvial system is highly stochastic and that the contribution of earthflow material of the last

[Figure]

~50 years makes up for more than half of the total sediment volume exported from the Entle catchment on average over millennial timescales.

Different techniques of measuring sediment fluxes allow us to estimate average fluxes exported from catchments over different timescales. Our knowledge on the variability of sediment production on hillslopes and its supply to river channels however is still limited. As such, I consider the manuscript of Clapuyt et al. as a valuable scientific contribution. While I appreciate the presented datasets and their comparison, I have two major concerns regarding (1) the analyses and interpretation of the 10Be data as well as (2) the presentation of the concepts. In addition, I raise a few minor concerns and provide further line-by-line comments, which are mainly related to the clarity of the manuscript and should be considered as suggestions. I suggest the manuscript for publication once the main concerns have been addressed.

Major comments

(1) The authors measure 10Be concentration in fluvial sediments, from which they calculate catchment average denudation rates as well as sediment fluxes by multiplying the denudation rates with the according catchment areas. When catchment-average denudation rates are calculated from detrital 10Be concentrations, one of the main assumptions is that each part of the catchment is equally represented in the sampled material. This assumption is violated when a sample is taken within or just downstream of a landslide deposit, because landslides are highly stochastic processes (as stated by the authors for example on p. 2 l. 3, p. 3 l. 10&12 or p. 13 l. 29). This is the case for the samples collected within the Schimbrig river. In such settings, the 10Be concentration in fluvial sediments collected at a certain moment in time is not necessarily representative of the long-term average and might be highly variable from year to year. Previous studies that have nicely demonstrated this are for example Dingle et al. (2018) or Lupker et al. (2012). For that reason, 10Be concentrations in fluvial sediments in landslide-prone areas are rather indicative of certain hillslope-erosion processes, but should be handled with care regarding the calculation of absolute values,

such as denudation rates or sediment fluxes. This problem also becomes apparent when the 4 data points from the Schimbrig catchment are compared with each other (Fig. 2). The last row within each box gives the calculated sediment flux (in volume per year). The sample located highest up within the catchment (CH-ENT-3) indicates a total annual sediment flux of 900 m3. When moving down the channel, the total annual sediment flux must increase, as the sediment discharge includes at least 900 m3 from the upstream part and additional sediment from the newly added catchment area. The values downstream, however, are about two thirds lower. As such, a reduction of sediment flux in downstream direction, despite total sediment flux being a cumulative parameter, clearly indicates a bias in the method. For the reasons listed above, I recommend the authors to be more careful with any of their mass-balance analyses that are based on calculated denudation rates and sediment fluxes from the landslide/ earthflow affected catchment. In particular, I disagree with the statement given for the temporal upscaling (section 4.1, p. 15 l. 8-9). The disagreement between decadal and millennial sediment fluxes can be purely a methodological problem. This also includes the comparison between the two Rossloch sub-catchments (p. 13 l. 7-11). The authors mention in their manuscript that also the gorge area is affected by landslides (p. 4 l. 27-29). Consequently, also the sample taken at the catchment outlet (E-7a) might be biased by mixing with low 10Be concentrations from landslide material. If so, the mass-balance exercise within the spatial upscaling (section 4.2., p. 15 l. 16-20) might also be biased. To address the above challenges, I suggest the authors to carefully re-evaluate their denudation rate and sediment flux analyses and interpretations and include a new section to the discussion that critically discusses the potential biases of the applied 10Be method and how this would affect their presented results.

(2) Secondly, I consider the discussion as largely under-cited. Although I really appreciate the detailed analysis of a single earthflow and the quantification of its contribution to the total sediment flux, the presented study is not the first study that has measured 10Be concentration in a landscape with stochastic sediment input, looked at evacuation timescales of stochastically supplied sediment or the potential alteration of sedimentary signals along sediment routing systems. None of the previous studies are cited in the discussion though. Rather, large parts of the discussion do not refer to any other studies at all. This includes most parts of the spatial upscaling (section 4.2) as well as large parts of the conceptual upscaling (section 4.3). To better highlight the novel findings of this work, the current study needs to be better embedded in the existing literature. A few suggestions for different topics are listed below, but many more are available. 10Be concentration in regions with stochastic sediment input: Puchol et al. (2014), Kober et al. (2012), West et al. (2014) Modification of sedimentary signals: van de Wiel and Coulthard (2010), Simpson and Castelltort (2012) Timescales of sediment removal provided by stochastic events: Hovius et al. (2000), Wang et al. (2015)

Minor comments

To better understand the novel contribution of the presented study, I suggest a clearer statement of the knowledge gap/ open question that is addressed by this work. In the current version the according statement within the abstract is rather vague (p. 2 l. 5-7). In the Introduction, the background knowledge is built up, but no clear research question is formulated. A good opportunity would be to insert a sentence on p.3 after line 25. Maybe it would also help to move this explaining sentence (p. 3 l. 27-29) further up before stating the question, as it can be seen as a motivation.

Please provide a more detailed characterization of the Schimbrig catchment, especially regarding the activity of hillslope processes apart from the earthflow itself (maybe add to p. 4 after l. 29). Could other processes within the catchment also affect the fluvial 10Be concentration? Along the same line, I would very much appreciate a photo of the Schimbrig earthflow.

p. 6 l. 18-26 and p. 8 l. 10-22: Please provide a more detailed explanation of decadal sediment flux method, as it is done for the other two methods. In particular, please indicate the areal extend covered by this methods (for example in figure 2). If I understand correctly, the annual analysis only covers the earthflow itself, while the decadal analysis

covers the entire catchment. To be able to compare the two, it would be interesting to know what other erosion processes are active in the catchment (see comment above) and what percentage of the catchment is affected/covered by the earth flow. Also, how is the displayed mass calculated (p. 8 l. 15-17)? I don't understand how this data is derived.

To ensure reproducibility of 10Be calculation and potential later re-analysis, please provide the raw data with the manuscript. This includes the original 10Be/9Be ratios from the AMS, as well as all the parameters needed to run the CAIRNs model. Also, was a correction for non-quartz containing areas within the catchments, as for example the carbonates, applied?

Line-by-line comments

p.3 l. 33-34: The sentence does not make sense as it is, please correct.

p. 4 l. 6-10: I suggest to number the analyses that are performed, as it makes it easier for the reader to follow the manuscript. However, I don't fully find the structure indicated here in the rest of the manuscript. Rather, the addressed topics are (i) temporal upscaling, (ii) spatial upscaling and (iii) conceptual upscaling. For clarification, I suggest to adapt this sentence, at least its order, or the way the data is later presented.

p. 4 l. 6: Inconsistent use of tenses, stick to one: 'discuss' is present tense, 'quantified' in past tense

p. 4 l. 16-19: As the 10Be concentration in fluvial quartz is measured later, it would help to provide information on the lithology/ quartz content in addition to the depositional types (molasse, flysch).

p. 4 l. 24-25: I don't follow the argument here. Why do differences in denudation rates point to a supply-limited system?

p. 6 l. 19: Is 'sediment yield' the same as 'sediment flux'? If so, consider changing it to flux to be consistent. Otherwise please define yield.

p. 6 l. 24: Was loose sediment or solid rock converted from tons per year into cubic meters per year? If it was converted from sediment, I would expect a lower density than 2.70 g/cm3.

p. 7 l. 1: In this sentence the authors state twice that their sample preparation was similar to other studies. What does 'similar' mean? Please be precise. Same accounts for the term 'several' in line 3.

p. 7 l. 7: Change 'is' to 'was' to be consistent in tenses.

p. 7 l. 27: What is meant by the term 'dynamic equilibrium'? Does it summarize what has been explained in the previous line, i.e. no net changes in volume? The way the sentence is written sounds to me like an interpretation, which would be miss-placed within the results sections.

p. 8 l. 22: I suggest to stick to one term, for instance earthflow when referring to the Schimbrig earthflow. In this sentence it is unclear if the 34 % come from the earthflow or also from other landslides that are active within the catchment? This is what motivated my comment above regarding a more detailed characterization of the hillslopes in the Schimbrig catchment.

p. 9 l. 15: It is unclear to which samples the term 'landslide-affected' refers to. For clarification, it would help to indicate in Table 3 which of the samples are considered as landslide-affected. I assume the term includes the 4 samples from the Schimbrig river. But why are 5 stars (= landslide-affected) displayed in the Fig. 3 and 4, but only 4 samples in that catchment? And is the Schimbrig earthflow the only landslide in the entire study-area, or could other samples also be considered as 'landslide-affected'?

p. 10 l. 6-7: I don't follow this interpretation. An increase in denudation rates in downstream direction could also be related to different local uplift rates, changes in lithology or recycling of the glacial till material (and as such not give 'true' denudation rates). Also, as this phrase is rather interpretation than a description of the results, the

authors could consider moving it to the 'Discussion' section of the manuscript.

p. 10 l. 6-7: I don't understand the sentence. What is meant by 'Accounting for the drainage area...'? Is the data displayed in Fig. 4 normalized by catchment area? If not (and it doesn't seem so), wouldn't an increase in sediment flux in downstream direction be expected as the sediment flux gives the total volume of sediment evacuated from a certain area per time? Consequently, the larger the area, the higher the sediment flux, even if denudation rates were constant across the entire area. Along the same line, I don't follow the statement on p. 12 l. 2-3.

p. 12 l. 16 - p. 13 l. 2: This sentence is rather discussion than a description of the results. Regarding its content, another possible explanation is that the fluvial sediments gets mixed with other, high 10Be sediment from within the catchment. This depends on what other processes are active within the catchment (see earlier comment).

p. 13 l. 6: Consider to also refer to Fig. 2 as this figure shows the variability in sediment fluxes across the entire study area.

p. 13 l. 20: km-2 yr-1, is that the correct unit?

p. 14 l. 18 – p. 15 l. 1: I suggest to replace 'the difference in denudations rates...' with 'the difference in 10Be concentration' as the denudation rates calculations are biased by the landslide and thus not reliable (see comment above).

p. 15 l. 2-3: What difference? The difference in sediment flux? And if it refers to the sediment flux, what about the other samples within the Schimbrig catchment? The uppermost sample (CH-ENT-3) already suggests an annual sediment evacuation of 900 m3, which is significantly higher than 230 m3 (CH-ENT-9). As such, I think the calculation of sediment flux from 10Be concentration in the earthflow affected catchments needs to be taken with care.

p. 15 l. 10: The importance OF landsliding...?

p. 15 l. 11-12: Or by a bias in the method, especially the 10Be derived sediment flux

calculations (see comments above).

p. 15 l. 19-20: If a mass-balance analysis is done, how about the other tributaries? If the contribution of all catchments is summed up, does it result in 100%?

p. 16 l. 6: Remove n from Entlen?

p.17 l. 15: 'pulses' instead of 'pulse'?

p.18 l. 11: Redistribution on the hillslopes, or just within the earthflow affected area? Please clarify.

p. 18 l. 21: Where does the 90% come from? Is this calculated from the data?

p. 18 l. 25-29: This statement is rather an interpretation about the evolution of such landscapes, which cannot directly be drawn from the presented data. Or if it can, I did not understand how it can be known from the presented dataset that once a sediment source is depleted, another landslide will be activated. Unless I missed something, I suggest reformulating the sentence to indicate it as an hypothesis that needs to be tested in the future.

Fig. 1: The elevation as supposedly shown in grayscale (legend) cannot be seen in the figure. I suggest to have two maps: one showing the DEM, and one showing the geological map. Maybe include a photo of the earth flow.

Fig. 3 and 4: The authors should consider to use different colors as red and green cannot be distinguished by a certain number of people.

References

Dingle, E. H., Sinclair, H. D., Attal, M., Rodés, Á., & Singh, V. (2018). Temporal variability in detrital Âź‰Be concentrations in a large Himalayan catchment. Earth Surface Dynamics, 6(3), 611-635.

Hovius, N., Stark, C.P., Hao‐Tsu, C., Jiun‐Chuan, L., 2000. Supply and Removal of Sediment in a Landslide‐Dominated Mountain Belt: Central Range, Taiwan. J. Geol. 108, 73–89. doi:10.1086/314387

Kober, F., Hippe, K., Salcher, B., Ivy-Ochs, S., Kubik, P.W., Wacker, L., Hählen, N., 2012. Debris-flow-dependent variation of cosmogenically derived catchment-wide denudation rates. Geology 40, 935–938. doi:10.1130/G33406.1

Lupker, M., Blard, P.-H., Lave, J., France-Lanord, C., Leanni, L., Puchol, N., Charreau, J., Bourlès, D., 2012. 10Be-derived Himalayan denudation rates and sediment budgets in the Ganga basin. Earth Planet. Sci. Lett. 333, 146–156.

Puchol, N., Lavé, J., Lupker, M., Blard, P., Gallo, F., France-Lanord, C., Team, A., 2014. Grain-size dependent concentration of cosmogenic 10Be and erosion dynamics in a landslide-dominated Himalayan watershed. Geomorphology 224, 55–68. doi:10.1016/j.geomorph.2014.06.019

Simpson, G., Castelltort, S., 2012. Model shows that rivers transmit high-frequency climate cycles to the sedimentary record. Geology 40, 1131–1134. doi:10.1130/G33451.1

Van de Wiel, M.J., Coulthard, T.J., 2010. Self-organized criticality in river basins: Challenging sedimentary records of environmental change. Geology 38, 87–90. doi:10.1130/G30490.1

Wang, J., Jin, Z., Hilton, R. G., Zhang, F., Densmore, A. L., Li, G., & West, A. J. (2015). Controls on fluvial evacuation of sediment from earthquake-triggered landslides. Geology, 43(2), 115-118

West, A.J., Hetzel, R., Li, G., Jin, Z., Zhang, F., Hilton, R.G., Densmore, A.L., 2014. Dilution of 10Be in detrital quartz by earthquake-induced landslides: Implications for determining denudation rates and potential to provide insights into landslide sediment dynamics. Earth Planet. Sci. Lett. 396, 143–153. doi:10.1016/j.epsl.2014.03.058

---

## Author Comment (AC1) · 16 May 2019

We would like to thank Luca Malatesta for the constructive comments and suggestions, which will guide us throughout the revision of the manuscript. Below, we respond to his suggestions and comments.

*Dear editor,*
*Clapuyt and colleagues present new constraints on geomorphic activity in an alpine catchment from cosmogenic radionuclides and from drone surveys (structure for motion). This new data complements existing constraints on sediment fluxes and allows a view across timescales from 10ˆ0 to 10ˆ3 yrs. The authors observe that the episodic activity of one earthflow leads to the production of a volume of sediment equivalent to that of the entire catchment, but that the evacuation of this material is limited by transport efficiency through the catchment such that earthflow activity is unlikely to imprint the stratigraphic record.*
*The science in this paper seems sound. I simply have one concern regarding potential recycling of glacial sediment for the cosmogenic radionuclide erosion rates. And I am confused by the interpretation of the catchment as a supply-limited vs. transport-limited system.*
*The language and the figures of the article are good. Though the dynamics of the earthflow are too summarily described and I have not really understood what is actually measured.*
*The manuscript, however, could be much improved by reworking its structure. At the moment, the novel contribution of the authors is somewhat buried under a discussion of known elements. I strongly encourage the authors to rethink the introduction and the motivation of their study to increase its impact. I provide some suggestions below.*
*All in all I recommend to accept this manuscript once the issue of potential sediment recycling is addressed and after 1) the sediment dynamics of the earth flow are more clearly defined and 2) the novel elements of the article are better highlighted.*

We take note of the recommendations and will address them carefully when revising our manuscript. Below, we provide a detailed reply to the issues raised.

*Sediment recycling: Figures 3 and 4 show a downstream increase in erosion rates and sediment fluxes once the Entle river flows in the inner gorge. The authors attribute this to the fast rate of postglacial incision in the gorge. To me, it however seems that the recycling of buried glacial sediment could be at least in part responsible for the trend of apparent increase in erosion rates caused by the increasing admixture of sediment with lower CRN concentration. If that is not a driver behind the increase in incision rate it should be explained. And if, on the contrary, this plays a role, this should be quantified.*

The Entle catchment was covered by glaciers during the LGM (24 ka BP, Bini et al., 2009). In the central part of the catchment, a 7 km-long inner gorge cuts through 100 m thick unconsolidated glacial deposits. The onset of incision of the inner gorge is still subject to debate. Van den Berg et al., (2012) argued that the incision age can be postglacial, given the high erodibility of the glacial material and high incision rate that might have operated during most of the knickzone propagation time.

Norton et al., (2008) showed that the [10]Be concentration of the glacial deposits in the Trub area (15 km to the W of our study site) equals $0.76 \pm 0.13 \times 10^4$ atoms g qtz$^{-1}$ at 8.5m depth. Also, they reported concentrations of $3.58 \pm 0.33 \times 10^4$ atoms g qtz$^{-1}$ at 1.5m depth, about 15%

to 50% higher than the catchment-wide CRN concentrations of nearby rivers. These data indicate that the decrease in $^{10}$Be along the Entle River (Fig. 3 & 4) is not necessarily linked to the incorporation of buried glacial material in the inner gorge.

In agreement with earlier work by Korup and Schlunegger (2007) and Van den Berg et al. (2012), we attribute the downstream decrease in $^{10}$Be concentrations to the contrasting geomorphic regimes between the inner gorge and the hillslopes above the knickzone.

We will discuss this point more extensively in the revised manuscript.

*Meaningfulness of the earthflow sediment dynamics: I do not think that I correctly understood what was being surveyed and what that entails for the sediment cascade. I have commented several parts of the manuscript (attached) where I might have been confused by a lack of clarity. Surface lowering on the earthflow is described as being the result of erosion (p. 6 l. 10-13). Isn't it also due to subsidence of the surface? I would expect erosion to mainly affect the bulging parts of the flow and mitigate the rate of surface uplift. Are subsidence and erosion differentiated? The lowering and rising parts of the flow both do so at the exact same mean rate. This is a rather arresting coincidence. It could be useful to add one sentence to explain/confirm this to avoid it being perceived as a red flag! The net mass flux of the earthflow is close to zero. This implies a constant volume though time. Then wouldn't the throughput flow, instead of the net balance, be the quantity that matters for sediment yield from the earthflow? Or alternatively, considering firstly the flux from bedrock to sediment (production) and secondly the loss of sediment from the earthflow to the channel (transfer). It would be potentially useful to reproduce a figure of deformation (bulging/lowering) on the earthflow in the manuscript to contextualize the values.*

Thank you for this comment. The paragraphs related to the annual sediment dynamics of the Schimbrig earthflow will be rephrased to clearly highlight what is measured and how we can further interpret these datasets. We will also add figures of the earthflow and its deformation, so that this part can stand alone, without making much reference to our previous work in Clapuyt et al. (2017). The sentence in p.6 l.10-13 stating that surface lowering is equivalent to erosion is an overstatement. Indeed, surface lowering does not necessarily mean erosion, especially in this context, as surface lowering occurs in the upper part of the earthflow while surface bulging occurs in the lower parts, i.e. highlighting the rotational structure of the mass movement. From the annual assessment of the earthflow sediment dynamics, we show that surface lowering is nearly equal to surface bulging. The mass balance of the earthflow is therefore roughly in equilibrium, meaning that the sediment mobilised by the earthflow is accumulating on the hillslope, indicating that the throughput flow is negligible during the short period 2013-2015. This is, however, not the case at the decadal scale (1962-1998) where Schwab et al. (2008) highlighted the very dynamic activity of the earthflow and quantified sediment fluxes to the river network.

*Sediment system: transport-limited or supply-limited? The catchment is framed as being supply-limited (p. 4 l. 24 and p. 17 l. 6-7). But it seems that the authors provide arguments for it being transport-limited at least in the first orders tributaries (p. 15 l. 25) the two conflicting accounts need to be reconciled. It is possible that a supply-limited catchment switches to being transport-limited when a landslide pulse overwhelms the transport paths.*

We agree with this comment. Both field observations and quantitative data presented in this research point to the hypothesis of a transport-limited drainage system. In the discussion, we argue for the capacity of the landscape to buffer stochastic sediment pulses from landslides. We will modify the text in this sense to gain in consistency.

*Structure: As it stands, I find that the article fails to properly motivate the study and to highlight the novelty the authors provide. In the discussion section, the authors use a significant amount of space to present already well-established conceptual models (sediment cascades, buffering of sediment pulses, stochasticity of landslides). The effect is to dilute the author's work. I believe it would be much more effective to introduce all these known/established elements at the beginning of the manuscript. This would allow the authors to explicitly define the gap in knowledge that their work directly addresses: a dataset across timescales, and not a conceptualisation of sediment transfer. I believe this would make it easier for the reader and increase the impact of the presented work. This section would also be the good place where to describe how the different processes affecting earthflow dynamics contribute to the sediment routing system.*

We will revise the structure of the manuscript according to your suggestion, which was implemented as is in a first draft of the manuscript. To address this comment, we will first clearly state the focus of our research in the introduction, i.e. the quantification of the propagation of stochastic sediment pulses across the landscape based on a multitemporal dataset of sediment fluxes. Then, we propose to distinguish the introduction section from the theoretical section about sediment transfer in landslide-prone environments, in order to keep the introduction quite short. In the discussion, we will only focus on our results and on what we can learn from integrating sediment fluxes over different time scales in a landslide-prone environment.

*I would like to encourage the authors to make better use of their data. Instead of synthetic data on the last figure, why don't they actually plot a distribution of erosion rates vs. timescale of integration (not time!) to present what is their truly significant contribution (data across timescales)? See Fig. 1 of Sadler 1981 for reference.*

We will add a figure in the discussion, depicting denudation rates over time scale of integration based on our data.

*The authors will find my line-by-line comments in the annotated pdf file hereby.*

We thank Luca for the thorough annotations of the manuscript, which will help to significantly improve its quality. We will address the line-by-line comments in the revised version.

*If any of my comments are unclear, the authors are welcome to contact me for clarification. Kind regards, Luca Malatesta*

**References**

Van den Berg, F., Schlunegger, F., Akçar, N. and Kubik, P.: 10Be-derived assessment of accelerated erosion in a glacially conditioned inner gorge, Entlebuch, Central Alps of Switzerland, Earth Surf. Process. Landforms, 37(11), 1176–1188, doi:10.1002/esp.3237, 2012.

Bini, A., Buonchristiani, J.-F., Couterand, S., Ellwanger, D., Felber, M., Florineth, D., Graf, H. R., Keller, O., Kelly, M., Schlüchter, C. and Schöneich, P.: Die Schweiz während des letzteiszeitlichen Maximums (LGM), 1:500 000, Bundesamt für Landestopographie, swisstopo. Wabern, Switz., 2009.

Clapuyt, F., Vanacker, V., Schlunegger, F. and Van Oost, K.: Unravelling earth flow dynamics with 3-D time series derived from UAV-SfM models, Earth Surf. Dyn., 5(4), 791–806, doi:10.5194/esurf-5-791-2017, 2017.

Korup, O. and Schlunegger, F.: Bedrock landsliding, river incision, and transience of geomorphic hillslope-channel coupling: Evidence from inner gorges in the Swiss Alps, J. Geophys. Res. Earth Surf., 112(3), doi:10.1029/2006JF000710, 2007.

Norton, K. P., von Blanckenburg, F., Schlunegger, F., Schwab, M. and Kubik, P. W.: Cosmogenic nuclide-based investigation of spatial erosion and hillslope channel coupling in the transient foreland of the Swiss Alps, Geomorphology, 95(3–4), 474–486, doi:10.1016/j.geomorph.2007.07.013, 2008.

Schwab, M., Rieke-Zapp, D., Schneider, H., Liniger, M. and Schlunegger, F.: Landsliding and sediment flux in the Central Swiss Alps: A photogrammetric study of the Schimbrig landslide, Entlebuch, Geomorphology, 97(3–4), 392–406, doi:10.1016/j.geomorph.2007.08.019, 2008.

---

## Author Comment (AC2) · 16 May 2019

We thank Reviewer 2 for his insightful and challenging comments, which will help us to significantly enhance the quality of our final manuscript.

*Dear Editor,*

*Clapuyt et al. investigate the sediment contribution of the Schimbrig earthflow (Switzerland) to the sediment flux of the entire Entle catchment with a particular focus of hillslope-channel coupling. They quantify the sediment flux of the earthflow on annual, decadal and millennial timescales by combining previously published data of sfm analysis of aerial photographs (annual) and time-series of photogrammetry-derived DEMs (decadal) with new and previously published 10Be-derived denudation rates (millennial). They conclude that sediment contribution from the earthflow to the fluvial system is highly stochastic and that the contribution of earthflow material of the last +/- 50 years makes up for more than half of the total sediment volume exported from the Entle catchment on average over millennial timescales.*

*Different techniques of measuring sediment fluxes allow us to estimate average fluxes exported from catchments over different timescales. Our knowledge on the variability of sediment production on hillslopes and its supply to river channels however is still limited. As such, I consider the manuscript of Clapuyt et al. as a valuable scientific contribution. While I appreciate the presented datasets and their comparison, I have two major concerns regarding (1) the analyses and interpretation of the 10Be data as well as (2) the presentation of the concepts. In addition, I raise a few minor concerns and provide further line-by-line comments, which are mainly related to the clarity of the manuscript and should be considered as suggestions. I suggest the manuscript for publication once the main concerns have been addressed.*

*Major comments*

*(1) The authors measure ¹⁰Be concentration in fluvial sediments, from which they calculate catchment average denudation rates as well as sediment fluxes by multiplying the denudation rates with the according catchment areas. When catchment-average denudation rates are calculated from detrital 10Be concentrations, one of the main assumptions is that each part of the catchment is equally represented in the sampled material. This assumption is violated when a sample is taken within or just downstream of a landslide deposit, because landslides are highly stochastic processes (as stated by the authors for example on p. 2 l. 3, p. 3 l. 10&12 or p. 13 l. 29). This is the case for the samples collected within the Schimbrig river. In such settings, the 10Be concentration in fluvial sediments collected at a certain moment in time is not necessarily representative of the long-term average and might be highly variable from year to year. Previous studies that have nicely demonstrated this are for example Dingle et al. (2018) or Lupker et al. (2012). For that reason, 10Be concentrations in fluvial sediments in landslide-prone areas are rather indicative of certain hillslope-erosion processes, but should be handled with care regarding the calculation of absolute values, such as denudation rates or sediment fluxes. This problem also becomes apparent when the 4 data points from the Schimbrig catchment are compared with each other (Fig. 2). The last row within each box gives the calculated sediment flux (in volume per year). The sample located highest up within the catchment (CH-ENT-3) indicates a total annual sediment flux of 900 m3. When moving down the channel, the total annual sediment flux must increase, as the sediment discharge includes at least 900 m3 from the upstream part and additional sediment from the newly added catchment area. The values downstream, however, are about two thirds*

*lower. As such, a reduction of sediment flux in downstream direction, despite total sediment flux being a cumulative parameter, clearly indicates a bias in the method. For the reasons listed above, I recommend the authors to be more careful with any of their mass-balance analyses that are based on calculated denudation rates and sediment fluxes from the landslide/ earthflow affected catchment. In particular, I disagree with the statement given for the temporal upscaling (section 4.1, p. 15 l. 8-9). The disagreement between decadal and millennial sediment fluxes can be purely a methodological problem. This also includes the comparison between the two Rossloch sub-catchments (p. 13 l. 7-11). The authors mention in their manuscript that also the gorge area is affected by landslides (p. 4 l. 27-29). Consequently, also the sample taken at the catchment outlet (E-7a) might be biased by mixing with low 10Be concentrations from landslide material. If so, the mass-balance exercise within the spatial upscaling (section 4.2., p. 15 l. 16-20) might also be biased. To address the above challenges, I suggest the authors to carefully re-evaluate their denudation rate and sediment flux analyses and interpretations and include a new section to the discussion that critically discusses the potential biases of the applied 10Be method and how this would affect their presented results.*

We acknowledge that landslides potentially dilute CRN concentrations and can introduce bias in the quantification of geomorphic processes. Therefore, in order to avoid overstatements using absolute values of denudation rates and sediment fluxes, we will discuss more extensively the $^{10}$Be concentrations as they were measured in river sediments of the Entle river catchment. If we consider $^{10}$Be concentrations and their potential dilution as a signal of landslides, our data clearly show that this effect is limited to the first-order Schimbrig catchment (Figure 1).

[Figure]

Figure 1: $^{10}$Be concentrations in the Entle catchment against downstream distance.

The $^{10}$Be concentration of sample CH-ENT-2 is in accordance with the decreasing linear trend when going downstream along the river network. Consequently, when entering the Kleine

Entle river, i.e. a second-order river, the signal of the landslide is not captured over a timescale of ca. 2,000 yr (Figure 2).

[Figure]

Figure 2: [10]Be concentrations in the Entle catchment against apparent age (yr).

However, as we are dealing with the sediment cascade of a mountainous environment, we do not agree with your statement that the total annual sediment flux must increase when going downstream, as it is a cumulative value. This is only true if the sediment transport rate is uniform over space and time. Mountainous river systems act as "jerky conveyor belts" (Ferguson, 1981) where sediment is transported episodically within catchments. Along the sediment cascade, sediment is sporadically deposited, eroded and transported, over different spatial and temporal scales (Fryirs, 2013). Here, the fact that the sediment flux is not cumulative when going downstream is precisely an indication of the capacity of the landscape to buffer sediment pulses, i.e. sediment mass from stochastic sediment pulses is trapped within first-order river catchments. These sediments are then progressively released further downstream over longer time scales. We see the same pattern in the landslide-affected catchment, over short time scales. A high landslide activity on the hillslopes does not necessarily lead to enhanced sediment fluxes at the outlet of the catchment.

In the inner gorge, we observe a decrease in [10]Be concentrations along the stream. This is related to deep seated landslides and gullies that are sourcing deeper material to the stream network. In the revised version of the paper, we will present the spatial variation in [10]Be concentrations in the area, and discuss potential caveats when deriving denudation rates from [10]Be concentrations in landslide-prone terrain.

*(2) Secondly, I consider the discussion as largely under-cited. Although I really appreciate the detailed analysis of a single earthflow and the quantification of its contribution to the total sediment flux, the presented study is not the first study that has measured 10Be concentration in a landscape with stochastic sediment input, looked at evacuation timescales of stochastically supplied sediment or the potential alteration of sedimentary signals along sediment routing systems. None of the previous studies are cited in the discussion though. Rather, large parts of the discussion do not refer to any other studies at all. This includes most parts of the spatial upscaling (section 4.2) as well as large parts of the conceptual*

*upscaling (section 4.3). To better highlight the novel findings of this work, the current study needs to be better embedded in the existing literature. A few suggestions for different topics are listed below, but many more are available. 10Be concentration in regions with stochastic sediment input: Puchol et al. (2014), Kober et al. (2012), West et al. (2014) Modification of sedimentary signals: van de Wiel and Coulthard (2010), Simpson and Castelltort (2012) Timescales of sediment removal provided by stochastic events: Hovius et al. (2000), Wang et al. (2015)*

We agree with this comment. We indeed missed to cite a series of papers dealing with the topic.  In the revised version, we will provide a thorough review of literature dealing with the integration of different timescales to assess the impact of landslides on the sediment cascade. This will help us to discuss and insert our results in a broader range of settings and environments.

*Minor comments*
*To better understand the novel contribution of the presented study, I suggest a clearer statement of the knowledge gap/ open question that is addressed by this work. In the current version the according statement within the abstract is rather vague (p. 2 l. 5- 7). In the Introduction, the background knowledge is built up, but no clear research question is formulated. A good opportunity would be to insert a sentence on p.3 after line 25. Maybe it would also help to move this explaining sentence (p. 3 l. 27-29) further up before stating the question, as it can be seen as a motivation.*

As also suggested by Reviewer 1, we will make the goal of the research more clear, at the end of the introduction, along with clearly distinguishing the current theoretical knowledge with respect to the discussion of our datasets. The open question that we want to quantify the propagation of stochastic sediment pulses throughout a mountainous drainage system using a multi-temporal approach and assess the capacity of the landscape to dampen these pulses in space and time.

*Please provide a more detailed characterization of the Schimbrig catchment, especially regarding the activity of hillslope processes apart from the earthflow itself (maybe add to p. 4 after l. 29). Could other processes within the catchment also affect the fluvial 10Be concentration? Along the same line, I would very much appreciate a photo of the Schimbrig earthflow. p. 6 l. 18-26 and p. 8 l. 10-22: Please provide a more detailed explanation of decadal sediment flux method, as it is done for the other two methods. In particular, please indicate the areal extend covered by this methods (for example in figure 2). If I understand correctly, the annual analysis only covers the earthflow itself, while the decadal analysis covers the entire catchment. To be able to compare the two, it would be interesting to know what other erosion processes are active in the catchment (see comment above) and what percentage of the catchment is affected/covered by the earth flow. Also, how is the displayed mass calculated (p. 8 l. 15-17)? I don't understand how this data is derived.*

We take note of this suggestion, and will add information on the hillslope processes active in the Schimbrig catchment. We will also include more detailed information on the methods that were used to derive the annual and decadal sediment fluxes, and provide a map with the outline of the datasets.

*To ensure reproducibility of 10Be calculation and potential later re-analysis, please provide the raw data with the manuscript. This includes the original 10Be/9Be ratios from the AMS, as well as all the parameters needed to run the CAIRNs model. Also, was a correction for non-quartz containing areas within the catchments, as for example the carbonates, applied?*

We will add this information on the derivation of [10]Be-denudation rates in the revised version of the manuscript.

*Line-by-line comments*

We thank the Reviewer 2 for the detailed line-by-line comments, which will enable us to improve the quality of the manuscript. We will address them in the revised version of the paper.

*p.3 l. 33-34: The sentence does not make sense as it is, please correct.*
*p. 4 l. 6-10: I suggest to number the analyses that are performed, as it makes it easier for the reader to follow the manuscript. However, I don't fully find the structure indicated here in the rest of the manuscript. Rather, the addressed topics are (i) temporal upscaling, (ii) spatial upscaling and (iii) conceptual upscaling. For clarification, I suggest to adapt this sentence, at least its order, or the way the data is later presented.*
*p. 4 l. 6: Inconsistent use of tenses, stick to one: 'discuss' is present tense, 'quantified' in past tense*
*p. 4 l. 16-19: As the 10Be concentration in fluvial quartz is measured later, it would help to provide information on the lithology/ quartz content in addition to the depositional types (molasse, flysch).*
*p. 4 l. 24-25: I don't follow the argument here. Why do differences in denudation rates point to a supply-limited system?*
*p. 6 l. 19: Is 'sediment yield' the same as 'sediment flux'? If so, consider changing it to flux to be consistent. Otherwise please define yield.*
*p. 6 l. 24: Was loose sediment or solid rock converted from tons per year into cubic meters per year? If it was converted from sediment, I would expect a lower density than 2.70 g/cm3.*
*p. 7 l. 1: In this sentence the authors state twice that their sample preparation was similar to other studies. What does 'similar' mean? Please be precise. Same accounts for the term 'several' in line 3.*
*p. 7 l. 7: Change 'is' to 'was' to be consistent in tenses.*
*p. 7 l. 27: What is meant by the term 'dynamic equilibrium'? Does it summarize what has been explained in the previous line, i.e. no net changes in volume? The way the sentence is written sounds to me like an interpretation, which would be miss-placed within the results sections.*
*p. 8 l. 22: I suggest to stick to one term, for instance earthflow when referring to the Schimbrig earthflow. In this sentence it is unclear if the 34%come from the earthflow or also from other landslides that are active within the catchment? This is what motivated my comment above regarding a more detailed characterization of the hillslopes in the Schimbrig catchment.*
*p. 9 l. 15: It is unclear to which samples the term 'landslide-affected' refers to. For clarification, it would help to indicate in Table 3 which of the samples are considered as landslide-affected. I assume the term includes the 4 samples from the Schimbrig river. But*

*why are 5 stars (= landslide-affected) displayed in the Fig. 3 and 4, but only 4 samples in that catchment? And is the Schimbrig earthflow the only landslide in the entire study-area, or could other samples also be considered as 'landslide-affected'?*

*p. 10 l. 6-7: I don't follow this interpretation. An increase in denudation rates in downstream direction could also be related to different local uplift rates, changes in lithology or recycling of the glacial till material (and as such not give 'true' denudation rates). Also, as this phrase is rather interpretation than a description of the results, the authors could consider moving it to the 'Discussion' section of the manuscript.*

*p. 10 l. 6-7: I don't understand the sentence. What is meant by 'Accounting for the drainage area: : :'? Is the data displayed in Fig. 4 normalized by catchment area? If not (and it doesn't seem so), wouldn't an increase in sediment flux in downstream direction be expected as the sediment flux gives the total volume of sediment evacuated from a certain area per time? Consequently, the larger the area, the higher the sediment flux, even if denudation rates were constant across the entire area. Along the same line, I don't follow the statement on p. 12 l. 2-3.*

*p. 12 l. 16 – p. 13 l. 2: This sentence is rather discussion than a description of the results. Regarding its content, another possible explanation is that the fluvial sediments gets mixed with other, high 10Be sediment from within the catchment. This depends on what other processes are active within the catchment (see earlier comment).*

*p. 13 l. 6: Consider to also refer to Fig. 2 as this figure shows the variability in sediment fluxes across the entire study area. p. 13 l. 20: km-2 yr-1, is that the correct unit?*

*p. 14 l. 18 – p. 15 l. 1: I suggest to replace 'the difference in denudations rates: : :' with 'the difference in 10Be concentration' as the denudation rates calculations are biased by the landslide and thus not reliable (see comment above).*

*p. 15 l. 2-3: What difference? The difference in sediment flux? And if it refers to the sediment flux, what about the other samples within the Schimbrig catchment? The uppermost sample (CH-ENT-3) already suggests an annual sediment evacuation of 900 m3, which is significantly higher than 230 m3 (CH-ENT-9). As such, I think the calculation of sediment flux from 10Be concentration in the earthflow affected catchments needs to be taken with care.*

*p. 15 l. 10: The importance OF landsliding: : :?*

*p. 15 l. 11-12: Or by a bias in the method, especially the 10Be derived sediment flux calculations (see comments above).*

*p. 15 l. 19-20: If a mass-balance analysis is done, how about the other tributaries? If the contribution of all catchments is summed up, does it result in 100%?*

*p. 16 l. 6: Remove n from Entlen?*

*p.17 l. 15: 'pulses' instead of 'pulse'?*

*p.18 l. 11: Redistribution on the hillslopes, or just within the earthflow affected area? Please clarify.*

*p. 18 l. 21: Where does the 90% come from? Is this calculated from the data?*

*p. 18 l. 25-29: This statement is rather an interpretation about the evolution of such landscapes, which cannot directly be drawn from the presented data. Or if it can, I did not understand how it can be known from the presented dataset that once a sediment source is depleted, another landslide will be activated. Unless I missed something, I suggest reformulating the sentence to indicate it as an hypothesis that needs to be tested in the future.*

*Fig. 1: The elevation as supposedly shown in grayscale (legend) cannot be seen in the figure. I suggest to have two maps: one showing the DEM, and one showing the geological map. Maybe include a photo of the earth flow. Fig. 3 and 4: The authors should consider to use different colors as red and green cannot be distinguished by a certain number of people.*

*References*

*Dingle, E. H., Sinclair, H. D., Attal, M., Rodés, Á., & Singh, V. (2018). Temporal variability in detrital 10Be concentrations in a large Himalayan catchment. Earth Surface Dynamics, 6(3), 611-635.*

*Hovius, N., Stark, C.P., HaoTsu, C., Jiun Chuan, L., 2000. Supply and Removal of Sediment in a LandslideDominated Mountain Belt: Central Range, Taiwan. J. Geol. 108, 73–89. doi:10.1086/314387*

*Kober, F., Hippe, K., Salcher, B., Ivy-Ochs, S., Kubik, P.W., Wacker, L., Hählen, N., 2012. Debris-flow-dependent variation of cosmogenically derived catchment-wide denudation rates. Geology 40, 935–938. doi:10.1130/G33406.1*

*Lupker, M., Blard, P.-H., Lave, J., France-Lanord, C., Leanni, L., Puchol, N., Charreau, J., Bourlès, D., 2012. 10Be-derived Himalayan denudation rates and sediment budgets in the Ganga basin. Earth Planet. Sci. Lett. 333, 146–156.*

*Puchol, N., Lavé, J., Lupker, M., Blard, P., Gallo, F., France-Lanord, C., Team, A., 2014. Grain-size dependent concentration of cosmogenic 10Be and erosion dynamics in a landslide-dominated Himalayan watershed. Geomorphology 224, 55–68. doi:10.1016/j.geomorph.2014.06.019*

*Simpson, G., Castelltort, S., 2012. Model shows that rivers transmit highfrequency climate cycles to the sedimentary record. Geology 40, 1131–1134. doi:10.1130/G33451.1*

*Van de Wiel, M.J., Coulthard, T.J., 2010. Self-organized criticality in river basins: Challenging sedimentary records of environmental change. Geology 38, 87–90. doi:10.1130/ G30490.1*

*Wang, J., Jin, Z., Hilton, R. G., Zhang, F., Densmore, A. L., Li, G., & West, A. J. (2015). Controls on fluvial evacuation of sediment from earthquake-triggered landslides. Geology, 43(2), 115-118*

*West, A.J., Hetzel, R., Li, G., Jin, Z., Zhang, F., Hilton, R.G., Densmore, A.L., 2014. Dilution of 10Be in detrital quartz by earthquake-induced landslides: Implications for determining denudation rates and potential to provide insights into landslide sediment dynamics. Earth Planet. Sci. Lett. 396, 143–153. doi:10.1016/j.epsl.2014.03.058*

**References**

Ferguson, R. I.: Channel forms and channel changes, in British Rivers, edited by L. J, pp. 90–125., 1981.

Fryirs, K.: (Dis)Connectivity in catchment sediment cascades: A fresh look at the sediment delivery problem, Earth Surf. Process. Landforms, 38(1), 30–46, doi:10.1002/esp.3242, 2013.

---

## Author Response (AR1)

**Letter to associate editor**

Dear associate editor,

5  We have revised our manuscript along the lines that the two reviewers suggested. In the letter below, you can find our point-by-point response to all comments. For the sake of clarity, the comments raised by each reviewer are copied in italic black font, and our response is given in blue below each comment. All changes were marked in yellow in the revised manuscript (below our response to reviewers), and we added the line numbers in our reply to the reviewers.

10  We want to take the opportunity to thank the reviewers for the thoughtful and constructive reviews that greatly contributed to improve the quality of our work.

We are looking forward to reading from you.

François Clapuyt, on behalf of the authors

Earth and Life Institute, UCLouvain

françois.clapuyt@uclouvain.be

**Reply to Reviewers.**

**Reviewer #1:**

*The science in this paper seems sound. I simply have one concern regarding potential recycling of glacial sediment for the cosmogenic radionuclide erosion rates. And I am confused by the interpretation of the catchment as a supply-limited vs. transport-limited system. The language and the figures of the article are good. Though the dynamics of the earthflow are too summarily described and I have not really understood what is actually measured. The manuscript, however, could be much improved by reworking its structure. At the moment, the novel contribution of the authors is somewhat buried under a discussion of known elements. I strongly encourage the authors to rethink the introduction and the motivation of their study to increase its impact. I provide some suggestions below. All in all I recommend to accept this manuscript once the issue of potential sediment recycling is addressed and after 1) the sediment dynamics of the earth flow are more clearly defined and 2) the novel elements of the article are better highlighted.*

We thank Luca Malatesta for his constructive comments. The recommendations helped us to better structure the manuscript and improve the quality of the science presented. Below, we provided a detailed reply to the issues raised.

*Sediment recycling: Figures 3 and 4 show a downstream increase in erosion rates and sediment fluxes once the Entle river flows in the inner gorge. The authors attribute this to the fast rate of postglacial incision in the gorge. To me, it however seems that the recycling of buried glacial sediment could be at least in part responsible for the trend of apparent increase in erosion rates caused by the increasing admixture of sediment with lower CRN concentration. If that is not a driver behind the increase in incision rate it should be explained. And if, on the contrary, this plays a role, this should be quantified.*

The Entle catchment was covered by glaciers during the LGM (24 ka BP, Bini et al., 2009). In the central part of the catchment, a 7 km-long inner gorge cuts through 100 m thick unconsolidated glacial deposits. The onset of incision of the inner gorge is still subject to debate. Van den Berg et al. (2012) argued that the incision age can be postglacial, given the high erodibility of the glacial material and high incision rate that might have operated during most of the knickzone propagation time.

Norton et al. (2008) showed that the [10]Be concentration of the glacial deposits in the Trub area (15 km to the W of our study site) equals $0.76 \pm 0.13 \times 10^4$ atoms g qtz$^{-1}$ at 8.5 m depth. Also, they reported concentrations of $3.58 \pm 0.33 \times 10^4$ atoms g qtz$^{-1}$ at 1.5 m depth, about 15% to 50% higher than the catchment-wide CRN concentrations of nearby rivers. These data indicate that the decrease in [10]Be along the Entle River (Figure 4; Figure 5) is not necessarily linked to the incorporation of buried glacial material in the inner gorge.

In agreement with earlier work by Korup and Schlunegger (2007) and Van den Berg et al. (2012), we attribute the downstream decrease in [10]Be concentrations to the contrasting geomorphic regimes between the inner gorge and the hillslopes above the knickzone. We have addressed this point in Sections 4.3 and 5.1.

*Meaningfulness of the earthflow sediment dynamics: I do not think that I correctly understood what was being surveyed and what that entails for the sediment cascade. I have commented several parts of the manuscript (attached) where I might have been confused by a lack of clarity. Surface lowering on the earthflow is described as being the result of erosion (p. 6 l. 10-13). Isn't it also due to subsidence of the surface? I would expect erosion to mainly affect the bulging parts of the flow and mitigate the rate of*

*surface uplift. Are subsidence and erosion differentiated? The lowering and rising parts of the flow both do so at the exact same mean rate. This is a rather arresting coincidence. It could be useful to add one sentence to explain/confirm this to avoid it being perceived as a red flag! The net mass flux of the earthflow is close to zero. This implies a constant volume though time. Then wouldn't the throughput flow, instead of the net balance, be the quantity that matters for sediment yield from the earthflow? Or alternatively, considering firstly the flux from bedrock to sediment (production) and secondly the loss of sediment from the earthflow to the channel (transfer). It would be potentially useful to reproduce a figure of deformation (bulging/lowering) on the earthflow in the manuscript to contextualize the values.*

Thank you for this comment. We revised the paragraphs related to the annual sediment dynamics of the Schimbrig earthflow, and clarified what is measured and how we interpreted the datasets. We added Figure 3 to illustrate the work done on the topographic reconstructions so that this part can stand alone without making much reference to our previous work in Clapuyt et al. (2017). The sentence in p.6 l.10-13 stating that surface lowering is equivalent to erosion is an overstatement. Indeed, surface lowering does not necessarily mean erosion, especially in this context, as surface lowering occurs in the upper part of the earthflow while surface bulging occurs in the lower parts, i.e. highlighting the rotational structure of the mass movement. From the annual assessment of the earthflow sediment dynamics, we show that surface lowering is nearly equal to surface bulging. The mass balance of the earthflow is therefore roughly in equilibrium, meaning that the sediment mobilised by the earthflow is accumulating on the hillslope, indicating that the throughput flow is negligible during the short period 2013-2015. This is, however, not the case at the decadal scale (1962-1998), as Schwab et al. (2008) showed that about 78% of the total mass displaced by the earthflow was evacuated by the channel network.

*Sediment system: transport-limited or supply-limited? The catchment is framed as being supply-limited (p. 4 l. 24 and p. 17 l. 6-7). But it seems that the authors provide arguments for it being transport-limited at least in the first orders tributaries (p. 15 l. 25) the two conflicting accounts need to be reconciled. It is possible that a supply-limited catchment switches to being transport-limited when a landslide pulse overwhelms the transport paths.*

We agree with this comment. Both field observations and quantitative data presented in this research point to the hypothesis of a transport-limited drainage system. In the discussion, we argue for the capacity of the landscape to buffer stochastic sediment pulses from landslides. We emphasised this point in the Discussion section in this sense to gain in consistency.

*Structure: As it stands, I find that the article fails to properly motivate the study and to highlight the novelty the authors provide. In the discussion section, the authors use a significant amount of space to present already well-established conceptual models (sediment cascades, buffering of sediment pulses, stochasticity of landslides). The effect is to dilute the author's work. I believe it would be much more effective to introduce all these known/established elements at the beginning of the manuscript. This would allow the authors to explicitly define the gap in knowledge that their work directly addresses: a dataset across timescales, and not a conceptualisation of sediment transfer. I believe this would make it easier for the reader and increase the impact of the presented work. This section would also be the good place where to describe how the different processes affecting earthflow dynamics contribute to the sediment routing system.*

We have revised the structure of the manuscript according to your suggestions. The last paragraph of the introduction was rewritten as to state the knowledge gap that we are addressing in the paper. Also, we followed up on your suggestion, and have reorganised the paper by inserting a section with the conceptual framework (Section 2) between the introduction and the methods. The discussion part was rewritten accordingly, to avoid repetition with the theoretical concepts presented in the conceptual framework.

*I would like to encourage the authors to make better use of their data. Instead of synthetic data on the last figure, why don't they actually plot a distribution of erosion rates vs. timescale of integration (not time!) to present what is their truly significant contribution (data across timescales)? See Fig. 1 of Sadler 1981 for reference.*

5    We updated Figure 7 (formerly Figure 4) in the discussion, depicting sediment fluxes over time scale of integration based on our data, i.e. at the millennial scale over the entire Entle catchment and at the decadal scale (black square markers) for the Schimbrig catchment.

**Line-by-line comments**

10  - *p. 3 l. 11: maybe define what the meaning/nature of that buffer is (sediment flux).*
      We specified it in the text p. 3 l. 14.

    - *p. 3 l. 23: no comma.*
      Comma removed.

    - *p. 4 l. 1: Few people will know right of the bat where the Entle River flows. Mention that it is in the*
15    *Swiss Alps. the earth flow should probably be introduced as "the Entle River catchment contains a large earthflow named Schimbrig." (I don't think that this earthflow has reached a level of fame warranting a "the").*
      We rephrased it accordingly p. 7 l. 4.

    - *p. 4 l. 5: it = sediment flux?*
20    Yes. We rephrased p. 5 l. 13-15.

    - *p. 4 l. 11: The largely review aspects of the discussion could be moved in a new section between intro and material. This section could end with a clear identification of a gap in knowledge motivating the current study. Thus the authors' work could more directly address a need. Effectively this would streamline the article*
25    We have carefully revised the structure of the article. We have added a separate section with the conceptual framework (Section 2), between the introduction and the method section. This allowed us to move part of the discussion to this new section. Also, we have clearly phrased the research questions in a new paragraph that was added at the end of the introduction, p. 3 l. 30-32 and p. 4 l. 1-5.

30  - *p. 4 l. 16: have Schlunegger et al. 2016a defined "Swiss Plateau"? Maybe this ref should cover the next sentence?*
      Correct, we have changed the location of the reference on p. 6 l. 9.

    - *p. 4 l. 21: in the figure 1 the gorge is mostly cut into the Molasse.*
      We updated Figure 2 with the updated version of the geological map, which shows a larger spatial
35    extent of glacial till deposits, corresponding to the sentence in the text p. 6 l. 9-11.

    - *p. 4 l. 23: erosion or incision? the latter would be the rate of lowering of the river itself. the former is ambiguous because we don't know over which area it is averaged.*
      It is incision indeed.
      We replaced "erosion" by "incision" p. 6 l. 15.

40  - *p. 4 l. 24: i don't follow the causality here. you haven't said anything about the transport regime or the upper reach geometry. for all i know the upper half of the catchment could be choked in sediment that are not being evacuated through the lower gorge. As a matter of fact, the poor efficiency in evacuating hillslope material suggests that there is a transport-limited dimension to the problem (p. 15, l. 21)*

We agree with the reviewer, and have adapted the text accordingly. The weak hillslope-channel coupling points to the existence of transport-limited systems.

- *p. 4 l. 26: "cut by" or "dissected by". to clarify that the moraine is not sitting next to and out of the gorge.*
We rephrased the idea p. 6 l. 12.

- *p. 4 l. 28: these areas are not "covered" by Flysch they are "made" of Flysch. Is there a reference for the abundance of earthflows in Flysch lithology?*
Sentence modified and reference added p. 6 l. 17-18.

- *p. 5 l. 1: I doubt that the satellite imagery would be useful when printed at this size. It might be better to have only political borders and main water bodies. potentially 1order tectonic boundaries (Alps - Foreland basin)*
Thanks for the comment. We modified the inset by including the first order geological setting and updated the extent of glacial deposits in panel (b) of Figure 2 (formerly Figure 1).

- *p. 5 l. 4: been particularly*
Modified in the text p. 7 l. 4.

- *p. 5 l. 7: is it esoterically connected to it then? consider replacing physically with directly.*
Modified in the text using the term directly p. 7 l. 8.

- *p. 5 l. 9: Reference for the description of the earthflow.*
Description of the earthflow is based on Schwab et al. (2008) and Clapuyt et al. (2017). References added p. 7 l. 10-11.

- *p. 6 l. 11: how are erosion and subsidence differentiated?*
Net erosion of the earthflow as a whole occurs when the balance between surface subsidence and bulging is positive, i.e. there is a positive sediment flux from the earthflow to the Schimbrig river. We rephrased and added information on that throughout Section 3.2.

- *p. 6 l. 12: Isn't it only the rate of erosion that reflects the flux of sediment provided to the system?*
Yes, we agree with your comment. It has been clarified at the beginning of Section 3.2 p. 8 l. 9-12.

- *p. 6 l. 17: reference?*
Reference on uncertainties and precision on SfM method, i.e. James et al. (2017), added p. 9 l. 9.

- *p. 6 l. 17: "spatializing the error" I don't understand what this means.*
It means that we used a spatially constant error value associated to topographic reconstructions. Precision made on p. 9 l. 8-9.

- *p. 6 l. 20: how were these loads measured?*
Sediment loads are derived from measurements of suspended sediment concentrations in a gauging station in the Waldemme River. We added the information in the text p. 9 l. 12-13.

- *p. 7 l. 19: several CRN samples lie downstream of glacial deposits. I expect these deposits to yield a significant fraction of the sediment flux at these locations. The denudation rates derived from samples with an important recycled component would have lower concentrations and thereby higher rates. How is that taken into account?*
We have addressed this point in Section 4.3. Data on [10]Be concentrations from glacial deposits published by Norton et al. (2008) support our argument that the decrease in [10]Be concentrations downstream is not primarily a result of admixture of glacial depostits.

- *p. 7 l. 22: Who is "we" Clapuyt et al. (2019) or Clapuyt et al. (2017), if it is the latter "they" would be clearer.*
It is Clapuyt et al. (2017). We changed "we" to "Clapuyt et al" on p. 10 l. 23.

- *p. 7 l. 22: i.e. which part of the flow?*
The active part of the earthflow surveyed by Clapuyt et al. (2017). We added the reference to Figure 4 where the extent is depicted.

- *p. 7 l. 22: across which gauge line is this flux constructed? i was expecting the sediment flux from the earth flow to correspond to its yield which has to be positive otherwise it would mean that sediment is climbing up from the river. so if this is flux corresponds to the overall budget of the active flow, it means that the flow has a stable volume through time. but for the question of sediment budget in the landscape, wouldn't the sediment throughput be the relevant value instead of the net balance?*
The gauging line is the entire hillslope affected by the earthflow and surrounded by stable pastures. As stated above, our annual assessment of the sediment dynamics show that surface lowering is nearly equal to surface bulging. The mass balance of the earthflow is therefore roughly in equilibrium, meaning that the sediment mobilised by the earthflow is accumulating on the hillslope, indicating that the throughput flow is negligible during the short period 2013-2015. We have clarified this in Section 3.2 and 3.3.

- *p. 7 l. 23: It is either positive, null, or negative. There are no qualification of how positive a number is. It is or it is not. it can be simpler to say that the flux "can be considered null"*
Thanks for the comment. We rephrased it on p. 10 l. 20-21.

- *p. 8 l. 3: should probably be m^2 only.*
Thanks for the comment. We changed it in Table 1.

- *p. 8 l. 3: it is not a depth but a rate*
Thanks for the comment. We changed it in Table 1.

- *p. 8 l. 16: what is the displaced mass? the throughput of the earth flow? needs more clarification about the origin of the percentage.*
We now give details on p. 9 l. 17-20 about the two variables that are computed by Schwab et al. (2008) in order to understand what is measured. The displaced mass corresponds to surface lowering while the total sediment exported per year, i.e. the sediment flux entering the Rossloch river, is the balance between surface lowering and bulging. The ratio between both metrics indicates the percentage of sediment mass evacuated compared to the displaced mass.

- *p. 8 l. 19: maybe add a column for the value of displaced mass (even though it is already folded into the percentage) that would allow to clearly explain what this value is.*
We added two columns in Table 2, i.e. total sediment displaced and exported (Schwab et al.,2008), in order to make everything clear and computable.

- *p. 10 l. 5: because of faster erosion or because of recycling?*
*p. 11 l. 1: higher rates due to recycling?*
*p. 12 l. 7: strong suggestion that increased admixture of recycled material yields faster rates.*
We refer to our reply about your main comment above (this issue is now addressed in the text in Sections 4.3 and 5.1)

- *p. 11 l. 2: precision: from CRN*
Precision done in the caption of Figure 6 (formerly Figure 3).

- *p. 12 l. 8: from CRN*
Precision done in the caption of Figure 7 (formerly Figure 4).

- *p. 13 l. 26: Has the toe of the landslide moved down at 6 m/yr? Or is the toe simply swelling and not moving?*
The entire earthflow is moving 6 m yr$^{-1}$ downslope on average. But the toe of the landslide only

experienced a downslope movement of ca. 55 m between 2014 and 2015. We rephrased on p. 18 l. 16-17.

[Figure]

- *p. 13 l. 29: there might be a need for precision here. shallow landslides are not "producing" sediment, they simply entrain them. Earthflows churn through bedrock effectively producing sediment. maybe this is actually a good way to frame the earthflow thoughput. I was confused by the meaningfulness of that value for sediment cascade but it could be defined in the intro that earthflows essentially produce colluvium/sediments. then question is what is the transport fate of that sediment.*

Thanks for the comment. The earthflow has deep rotational structure that actually produce sediment by excavating and mobilizing them. It therefore acts as a pure sediment source. We emphasised the fact that the earthflow acts as a sediment factory in Section 3.1 on p. 7 l. 4-6. And we added the info on the landslide toe advance on p. 18 l. 16-17.

- *p. 13 l. 31: Where does the rest remain? In which form does it remain on the hill slope.*

It is stored as colluvial fans at the foot of the earthflow, i.e. Figure 3 in Section 3.2. The sentence has been removed when reshaping the discussion.

- *p. 14 l. 3: this mechanism should be introduced earlier when the signification of an earthflow for the sediment cascade is explained.*

Thanks for the comment. We moved this part after the introduction, in the "conceptual framework" section (Section 2).

- *p. 14 l. 6: All the paragraph: this type of information could be used at the beginning for a motivation of the study. it can help defining a knowledge gap.*

Thanks for the comment. We moved this part after the introduction, in the "conceptual framework" section (Section 2).

- *p. 14 l. 11: can the actual data be plotted this way?*

See next comment.

- *p. 14 l. 11: three timescales are compared here using mock-data. Wouldn't it be possible to plot the actual data produced and compiled in this manuscript to test the dependency of observed rate on the time window of measurement (Fig. 1 of Sadler, 1981). Or to complement this conceptual graph with the actual data in a plot next to it.*

We have not enough data in this paper to implement a figure such as the one of Sadler (1981). However, we updated Figure 7 (formerly Figure 4) the sediment fluxes computed at the decadal scale, to enable the comparison for the Schimbrig catchment. We now emphasize in the text that Figure 8 is an extrapolation of our findings, in accordance with other studies about evacuation of sediments in landslide-prone environments.

- *p. 15 l. 2: What is compared here? total erosion rates? total denudation rate, total sediment flux?*
  We compare sediment fluxes. We rephrased on p. 18 l. 1-7.

- *p. 15 l. 9: the argument would be compelling if the actual data was plotted.*
  Thanks for the comment. We plotted it in Figure 7.

- *p. 15 l. 21: unclear. Because the system is transport-limited, or because the sediment pulse is diluted in a larger flux?*
  We have rephrased this sentence for clarification.

- *p. 15 l. 24: This seems to be an important element of the study. The upper reaches seem to be transport-limited. This is in part conflicting with an earlier statement about supply-limited upper reaches (p. 4 l. 24).*
  We rephrased the text in Section 3.1 as it was inaccurate. We agree with your observation that the upper reaches of the Entle catchment are transport-limited, as it is indeed suggested by our spatio-temporal database of sediment fluxes.

- *p. 15 l. 27: is it though? is there such a thing as a state of equilibrium in a situation like this? maybe when the earth flow scarp reaches the ridge and the slope can be in equilibrium again?*
  We agree with your statement, and have rephrased this part.

- *p. 15 l. 30: It might be opportune to frame it as a transport-limited system that delays and dilutes the propagation of the pulse*
  Done p. 19 l. 1-5 and p. 20 l. 4-6.

- *p. 16 l. 9: this again could be used in the intro to motivate the study. this is not inherently new stuff.*
  This sentence has been removed when restructuring the discussion and separating current knowledge and new findings of this paper.

- *p. 17 l. 5: but interestingly you have repeat observations suggesting that the system is transport-limited (p. 15, l. 21) elaborate!*
  We reworked the discussion part, as there was some repetitions between the different sections. We elaborated the idea of the transport-limited systems, and the geomorphic decoupling of slopes and channels in Sections 5.2 and 5.3.

- *p. 17 l. 10: of the hillslope or of the catchment?*
  Both on hillslopes and in the catchment. However, this sentence has been removed when restructuring the discussion and separating current knowledge and new findings of this paper.

- *p. 17 l. 17: ok but this is not new. what is new is that quantitative constraints are produced*

- We added our data in this section, to show the quantitative estimates of the sediment fluxes at the different spatial and temporal scales

- *p. 18 l. 13: average of what?*
  *p. 18 l. 15: More so than if it was low magnitude, high frequency?*
  *p. 18 l. 22: does that mean that the total Qs doubles, or that the provenance fraction of a constant total Qs changes? If the total Qs does not change there is an interesting observation of transport-limited rates of sediment evacuation at the scale of the catchment.*
  *p. 18 l. 26: causality unclear to me. Why would the another catchment be affected ONCE the previous one ceases its activity?*
  We have rephrased this part of the discussion, taking into account the previous comments that were related to poor wording.

**Reviewer #2:**

*Different techniques of measuring sediment fluxes allow us to estimate average fluxes exported from catchments over different timescales. Our knowledge on the variability of sediment production on hillslopes and its supply to river channels however is still limited. As such, I consider the manuscript of Clapuyt et al. as a valuable scientific contribution. While I appreciate the presented datasets and their comparison, I have two major concerns regarding (1) the analyses and interpretation of the 10Be data as well as (2) the presentation of the concepts. In addition, I raise a few minor concerns and provide further line-by-line comments, which are mainly related to the clarity of the manuscript and should be considered as suggestions. I suggest the manuscript for publication once the main concerns have been addressed.*

We thank Reviewer 2 for the challenging comments that helped us to improve the manuscript. Below, we provided a detailed reply to the issues raised.

*Major comments*

*(1) The authors measure 10Be concentration in fluvial sediments, from which they calculate catchment average denudation rates as well as sediment fluxes by multiplying the denudation rates with the according catchment areas. When catchment-average denudation rates are calculated from detrital 10Be concentrations, one of the main assumptions is that each part of the catchment is equally represented in the sampled material. This assumption is violated when a sample is taken within or just downstream of a landslide deposit, because landslides are highly stochastic processes (as stated by the authors for example on p. 2 l. 3, p. 3 l. 10&12 or p. 13 l. 29). This is the case for the samples collected within the Schimbrig river. In such settings, the 10Be concentration in fluvial sediments collected at a certain moment in time is not necessarily representative of the long-term average and might be highly variable from year to year. Previous studies that have nicely demonstrated this are for example Dingle et al. (2018) or Lupker et al. (2012). For that reason, 10Be concentrations in fluvial sediments in landslide-prone areas are rather indicative of certain hillslope-erosion processes, but should be handled with care regarding the calculation of absolute values, such as denudation rates or sediment fluxes. This problem also becomes apparent when the 4 data points from the Schimbrig catchment are compared with each other (Fig. 2). The last row within each box gives the calculated sediment flux (in volume per year). The sample located highest up within the catchment (CH-ENT-3) indicates a total annual sediment flux of 900 m3. When moving down the channel, the total annual sediment flux must increase, as the sediment discharge includes at least 900 m3 from the upstream part and additional sediment from the newly added catchment area. The values downstream, however, are about two thirds lower. As such, a reduction of sediment flux in downstream direction, despite total sediment flux being a cumulative parameter, clearly indicates a bias in the method. For the reasons listed above, I recommend the authors to be more careful with any of their mass-balance analyses that are based on calculated denudation rates and sediment fluxes from the landslide/ earthflow affected catchment. In particular, I disagree with the statement given for the temporal upscaling (section 4.1, p. 15 l. 8-9). The disagreement between decadal and millennial sediment fluxes can be purely a methodological problem. This also includes the comparison between the two Rossloch sub-catchments (p. 13 l. 7-11). The authors mention in their manuscript that also the gorge area is affected by landslides (p. 4 l. 27-29). Consequently, also the sample taken at the catchment outlet (E-7a) might be biased by mixing with low 10Be concentrations from landslide material. If so, the mass-balance exercise within the spatial upscaling (section 4.2., p. 15 l. 16-20) might also be biased. To address the above challenges, I suggest the authors to carefully re-evaluate their denudation rate and sediment flux analyses and interpretations and include a new section to the discussion that critically discusses the potential biases of the applied 10Be method and how this would affect their presented results.*

We acknowledge that landslides potentially dilute CRN concentrations and can introduce bias in the quantification of geomorphic processes. Therefore, in order to avoid overstatements using absolute values of denudation rates and sediment fluxes, we commented [10]Be concentrations only in the Results section, i.e. Section 4. Then, we opened the discussion section, i.e. Section 5.1, by acknowledging three potential caveats using [10]Be concentrations to quantify derived denudation rates and sediment fluxes in such geomorphologic settings, i.e. glacial sediment admixture and dilution of CRN concentrations. Regarding the latter caveat, we argue that dilution lead to a potential overestimation of CRN-derived denuation rates (see p. 14 l. 13-18 and p. 15 l. 1-16). Therefore, CRN-derived denudation rates and subsequent sediment fluxes presented hereunder should be taken as first order or maximum estimates of the actual values. Because the decadal sediment flux from the Schimbrig catchment is two order of magnitude higher than the one computed at the millennial scale, this overestimation does not eventually affect the conclusions drawn from the multi-temporal database of sediment fluxes.

The [10]Be concentration of sample CH-ENT-2 is in accordance with the decreasing linear trend when going downstream along the river network. Consequently, when entering the Kleine Entle river, i.e. a second-order river, the signal of the landslide is not captured over a timescale of ca. 2,000 yr. We added the apparent age of [10]Be measurements in Table 3.

However, as we are dealing with the sediment cascade of a mountainous environment, we do not necessarily agree with your statement that the total annual sediment flux must increase when going downstream, as it is a cumulative value. This is only true if the sediment transport rate is uniform over space and time. Mountainous river systems act as "jerky conveyor belts" (Ferguson, 1981) where sediment is transported episodically within catchments. Along the sediment cascade, sediment is sporadically deposited, eroded and transported, over different spatial and temporal scales (Fryirs, 2013). Here, the fact that the sediment flux is not cumulative when going downstream is precisely an indication of the capacity of the landscape to buffer sediment pulses, i.e. sediment mass from stochastic sediment pulses is trapped within first-order river catchments. These sediments are then progressively released further downstream over longer time scales. We see the same pattern in the landslide-affected catchment, over short time scales. A high landslide activity on the hillslopes does not necessarily lead to enhanced sediment fluxes at the outlet of the catchment.

*(2) Secondly, I consider the discussion as largely under-cited. Although I really appreciate the detailed analysis of a single earthflow and the quantification of its contribution to the total sediment flux, the presented study is not the first study that has measured 10Be concentration in a landscape with stochastic sediment input, looked at evacuation timescales of stochastically supplied sediment or the potential alteration of sedimentary signals along sediment routing systems. None of the previous studies are cited in the discussion though. Rather, large parts of the discussion do not refer to any other studies at all. This includes most parts of the spatial upscaling (section 4.2) as well as large parts of the conceptual upscaling (section 4.3). To better highlight the novel findings of this work, the current study needs to be better embedded in the existing literature. A few suggestions for different topics are listed below, but many more are available. 10Be concentration in regions with stochastic sediment input: Puchol et al. (2014), Kober et al. (2012), West et al. (2014) Modification of sedimentary signals: van de Wiel and Coulthard (2010), Simpson and Castelltort (2012) Timescales of sediment removal provided by stochastic events: Hovius et al. (2000), Wang et al. (2015)*

We agree with this comment. We indeed missed to cite a series of papers dealing with the topic. We added references to relevant literature in the discussion section.

*Minor comments*

*To better understand the novel contribution of the presented study, I suggest a clearer statement of the knowledge gap/ open question that is addressed by this work. In the current version the according statement within the abstract is rather vague (p. 2 l. 5- 7). In the Introduction, the background knowledge*

*is built up, but no clear research question is formulated. A good opportunity would be to insert a sentence on p.3 after line 25. Maybe it would also help to move this explaining sentence (p. 3 l. 27-29) further up before stating the question, as it can be seen as a motivation.*

As also suggested by Reviewer 1, we have rewritten the last part of the introduction where we now introduce the research hypothesis that is addressed in our paper (p. 3 l. 30-32 and p. 4 l. 1-5): "In this study, we posit that sediment fluxes in landslide-prone alpine catchments can be highly variable in space and time, with long periods of quiescence during which sediment is temporarily stored on the hillslopes and short episodes of high sediment flux when hillslopes and channel are coupled through superimposed debris flows".

*Please provide a more detailed characterization of the Schimbrig catchment, especially regarding the activity of hillslope processes apart from the earthflow itself (maybe add to p. 4 after l. 29). Could other processes within the catchment also affect the fluvial 10Be concentration? Along the same line, I would very much appreciate a photo of the Schimbrig earthflow. p. 6 l. 18-26 and p. 8 l. 10-22: Please provide a more detailed explanation of decadal sediment flux method, as it is done for the other two methods. In particular, please indicate the areal extend covered by this methods (for example in figure 2). If I understand correctly, the annual analysis only covers the earthflow itself, while the decadal analysis covers the entire catchment. To be able to compare the two, it would be interesting to know what other erosion processes are active in the catchment (see comment above) and what percentage of the catchment is affected/covered by the earth flow. Also, how is the displayed mass calculated (p. 8 l. 15-17)? I don't understand how this data is derived.*

We added information on the different types of erosion processes in the Entle and Schimbrig catchments, respectively p. 6 l. 15-20 and p. 8 l. 1-4. We added pictures of the earthflow in Figure 3.

Regarding decadal sediment fluxes computation, we detailed the workflow and detailed how metrics presented in Table 2 are computed on p. 9 l. 17-20. In Figure 4 (formerly Figure 2), we added the outline of the Schimbrig catchment over which the decadal sediment fluxes are computed.

We added the importance of the Schimbrig earthflow within the catchment in Section 3.1 on p. 8 l. 3-4, i.e. 25% of the Schimbrig catchment in surface.

*To ensure reproducibility of 10Be calculation and potential later re-analysis, please provide the raw data with the manuscript. This includes the original 10Be/9Be ratios from the AMS, as well as all the parameters needed to run the CAIRNs model. Also, was a correction for non-quartz containing areas within the catchments, as for example the carbonates, applied?*

We added original measured $^{10}Be/^{9}Be$ ratios from the AMS in Table 3.

**Line-by-line comments**

- *p.3 l. 33-34: The sentence does not make sense as it is, please correct.*
  Sentence corrected p. 5 l. 18-19.

- *p. 4 l. 6-10: I suggest to number the analyses that are performed, as it makes it easier for the reader to follow the manuscript. However, I don't fully find the structure indicated here in the rest of the manuscript. Rather, the addressed topics are (i) temporal upscaling, (ii) spatial upscaling and (iii) conceptual upscaling. For clarification, I suggest to adapt this sentence, at least its order, or the way the data is later presented.*
  We restructured the entire paragraph to better highlight the focus of the paper.

- *p. 4 l. 6: Inconsistent use of tenses, stick to one: 'discuss' is present tense, 'quantified' in past tense.*
  Section rephrased but now tenses are correct.

- *p. 4 l. 16-19: As the 10Be concentration in fluvial quartz is measured later, it would help to provide information on the lithology/ quartz content in addition to the depositional types (molasse, flysch).*
  We added this information in the methodology section about CRN analyses p. 10 l. 3-4.

- *p. 4 l. 24-25: I don't follow the argument here. Why do differences in denudation rates point to a supply-limited system?*
  We have rephrased this part to avoid confusion.

- *p. 6 l. 19: Is 'sediment yield' the same as 'sediment flux'? If so, consider changing it to flux to be consistent. Otherwise please define yield.*
  We have rephrased this sentence: "… who assessed sediment transport by…"

- *p. 6 l. 24: Was loose sediment or solid rock converted from tons per year into cubic meters per year? If it was converted from sediment, I would expect a lower density than 2.70 g/cm3.*
  We ignore the potential volumetric expansion of the earthflow material as we are interested in the general pattern of geomorphic responses rather than in the absolute magnitude of the responses.

- *p. 7 l. 1: In this sentence the authors state twice that their sample preparation was similar to other studies. What does 'similar' mean? Please be precise. Same accounts for the term 'several' in line 3.*
  We meant that the protocol followed is described in Vanacker et al. (2007) and has also been followed by Van den Berg et al. (2012). We rephrased on p. 9 l. 28-29.
  We replaced "several" by "up to 10" on p. 9 l. 31 as it is varying according to samples.

- *p. 7 l. 7: Change 'is' to 'was' to be consistent in tenses.*
  Correction done p. 10 l. 2.

- *p. 7 l. 27: What is meant by the term 'dynamic equilibrium'? Does it summarize what has been explained in the previous line, i.e. no net changes in volume? The way the sentence is written sounds to me like an interpretation, which would be miss-placed within the results sections.*
  "Dynamic equilibrium" reflects the fact that there is no net flux to the river, only a redistribution of sediments on the hillslope or a balance between surface lowering and bulging. To us, it is rather a summary of what is previously written than an interpretation that should go the discussion section.

- *p. 8 l. 22: I suggest to stick to one term, for instance earthflow when referring to the Schimbrig earthflow. In this sentence it is unclear if the 34%come from the earthflow or also from other landslides that are active within the catchment? This is what motivated my comment above regarding a more detailed characterization of the hillslopes in the Schimbrig catchment.*
  You are right about vocabulary consistency. We changed "landslide" to "earthflow" on p. 11 l. 14 and checked the entire document for similar mistakes.

The 34% of displaced material accounts only for the Schimbrig earthflow. We did not focus on other landslides in the Entle catchment, as stated in the Materiel and Methods section. Following your previous comment, we added more details on the erosion processes in Section 3.1 p. 8 l. 1-4.

- *p. 9 l. 15: It is unclear to which samples the term 'landslide-affected' refers to. For clarification, it would help to indicate in Table 3 which of the samples are considered as landslide-affected. I assume the term includes the 4 samples from the Schimbrig river. But why are 5 stars (= landslide-affected) displayed in the Fig. 3 and 4, but only 4 samples in that catchment? And is the Schimbrig earthflow the only landslide in the entire study-area, or could other samples also be considered as 'landslide-affected'?*

Thanks for the comment. One star symbol in Figures 6 and 7 (formerly Figure 3 and 4) corresponded to knickpoint locations. We changed this symbol to a triangle to avoid confusion. We make the hypothesis that the Schimbrig catchment only affects the CRN concentrations of samples.

- *p. 10 l. 6-7: I don't follow this interpretation. An increase in denudation rates in downstream direction could also be related to different local uplift rates, changes in lithology or recycling of the glacial till material (and as such not give 'true' denudation rates). Also, as this phrase is rather interpretation than a description of the results, the authors could consider moving it to the 'Discussion' section of the manuscript.*
We moved this interpretation into Section 5.1.

- *p. 10 l. 6-7: I don't understand the sentence. What is meant by 'Accounting for the drainage area: : :'? Is the data displayed in Fig. 4 normalized by catchment area? If not (and it doesn't seem so), wouldn't an increase in sediment flux in downstream direction be expected as the sediment flux gives the total volume of sediment evacuated from a certain area per time? Consequently, the larger the area, the higher the sediment flux, even if denudation rates were constant across the entire area. Along the same line, I don't follow the statement on p. 12 l. 2-3.*
"Accounting for drainage area" means that we deal with sediment fluxes instead of denudation rates. We precised it on p. 15 l. 22-23. In Figure 7 (formerly Figure 4), we plot sediment fluxes ($L^3$ $T^{-1}$), i.e. the total volume of sediment evacuated from a catchment, which increase downstream. This actually supports your comment.
We agree with you that the increase in sediment fluxes along the river network indicates that sediments are evactuated from the system. Our point is in fact that in the landslide-affected catchment, the opposite trend is observed, meaning that sediments are temporarily stored or deposited in this first-order catchment (see p. 17 l. 1-11), i.e. illustrating the buffering capacity of the landscape facing stochastic sediment pulses.

- *p. 12 l. 16 – p. 13 l. 2: This sentence is rather discussion than a description of the results. Regarding its content, another possible explanation is that the fluvial sediments gets mixed with other, high 10Be sediment from within the catchment. This depends on what other processes are active within the catchment (see earlier comment).*
We agree with your comment and moved this interpretation to Section 5.1 (see p.15 l. 5-11). Besides, as precised in Section 3.1 (p. 8 l. 3-4), the Schimbrig earthflow occupies 25% of the catchment area and is the only active process.

- *p. 13 l. 6: Consider to also refer to Fig. 2 as this figure shows the variability in sediment fluxes across the entire study area. p. 13 l. 20: km-2 yr-1, is that the correct unit?*
We added a reference to Figure 4 (formerly Figure 2) on p. 17 l. 14. It is the correct unit, i.e. the landslide rate as a number of landslides per square kilometre per year (see e.g. Hovius et al., 2000). To remove all doubts, we changed it to *events $km^{-2}$ $yr^{-1}$*. See p. 18 l. 16.

- *p. 14 l. 18 – p. 15 l. 1: I suggest to replace 'the difference in denudations rates: : :' with 'the difference in 10Be concentration' as the denudation rates calculations are biased by the landslide and thus not reliable (see comment above).*
  As answered to your first main comment, we moved these interpretations to Section 5.1, after discussing their robustness.

- *p. 15 l. 2-3: What difference? The difference in sediment flux? And if it refers to the sediment flux, what about the other samples within the Schimbrig catchment? The uppermost sample (CH-ENT-3) already suggests an annual sediment evacuation of 900 m3, which is significantly higher than 230 m3 (CH-ENT-9). As such, I think the calculation of sediment flux from 10Be concentration in the earthflow affected catchments needs to be taken with care.*
  We refer to our detailed reply to major comment (1) above.

- *p. 15 l. 10: The importance OF landsliding: ?*
  We rephrased the title of Section 5.2 as "The stochastic nature of landsliding".

- *p. 15 l. 11-12: Or by a bias in the method, especially the 10Be derived sediment flux calculations (see comments above).*
  We refer to our detailed reply to major comment (1) above.

- *p. 15 l. 19-20: If a mass-balance analysis is done, how about the other tributaries? If the contribution of all catchments is summed up, does it result in 100%?*
  We refer to our reply to major comment (1) above.

- *p. 16 l. 6: Remove n from Entlen?*
  Removed as we restructured the discussion.

- *p.17 l. 15: 'pulses' instead of 'pulse'?*
  Not applicable anymore due to rephrasing.

- *p.18 l. 11: Redistribution on the hillslopes, or just within the earthflow affected area? Please clarify.*
  We have rephrased this sentence.

- *p. 18 l. 21: Where does the 90% come from? Is this calculated from the data?*
  Thank you for the comment. There was a mistake, and we made the necessary corrections in Section 5.3.

- *p. 18 l. 25-29: This statement is rather an interpretation about the evolution of such landscapes, which cannot directly be drawn from the presented data. Or if it can, I did not understand how it can be known from the presented dataset that once a sediment source is depleted, another landslide will be activated. Unless I missed something, I suggest reformulating the sentence to indicate it as an hypothesis that needs to be tested in the future.*
  We agree with the comment and rephrased the last part of the conclusion.

- *Fig. 1: The elevation as supposedly shown in grayscale (legend) cannot be seen in the figure. I suggest to have two maps: one showing the DEM, and one showing the geological map. Maybe include a photo of the earth flow. Fig. 3 and 4: The authors should consider to use different colors as red and green cannot be distinguished by a certain number of people.*
  Regarding the elevation displayed as grayscale on Figure 1, we agree that the superimposed hillshade nearly entirely overrides the DEM. We keep the scale bar in the legend to give an indication about the elevation range. Nevertheless, we do not think that a map displaying the DEM only is very relevant.
  Thank you for the comment about the readability of the figures by color-blind people. We reviewed

all the figures of the manuscript and adapted them accordingly.

[revised manuscript text omitted]